# The Bayesian sampling in a canonical recurrent circuit with a diversity of inhibitory interneurons

**Eryn Sale**[1,2]
eryn.sale@utsouthwestern.edu

**Wen-Hao Zhang**[1,2*]
wenhao.zhang@utsouthwestern.edu

[1]Lyda Hill Department of Bioinformatics, UT Southwestern Medical Center
[2]O'Donnell Brain Institute, UT Southwestern Medical Center

## Abstract

Accumulating evidence suggests stochastic cortical circuits can perform sampling-based Bayesian inference to compute the latent stimulus posterior. Canonical cortical circuits consist of excitatory (E) neurons and types of inhibitory (I) interneurons. Nevertheless, nearly no sampling neural circuit models consider the diversity of interneurons, and thus how interneurons contribute to sampling remains poorly understood. To provide theoretical insight, we build a *nonlinear* canonical circuit model consisting of recurrently connected E neurons and two types of I neurons including Parvalbumin (PV) and Somatostatin (SOM) neurons. The E neurons are modeled as a canonical ring (attractor) model, receiving global inhibition from PV neurons, and locally tuning-dependent inhibition from SOM neurons. We theoretically analyze the *nonlinear* circuit dynamics and *analytically* identify the Bayesian sampling algorithm performed by the circuit dynamics. We found a reduced circuit with only E and PV neurons performs Langevin sampling, and the inclusion of SOM neurons with tuning-dependent inhibition speeds up the sampling via upgrading the Langevin into Hamiltonian sampling. Moreover, the Hamiltonian framework requires SOM neurons to receive no direct feedforward connections, consistent with neuroanatomy. Our work provides overarching connections between nonlinear circuits with various types of interneurons and sampling algorithms, deepening our understanding of circuit implementation of Bayesian inference.

## 1 Introduction

The brain lives in a world of uncertainty and ambiguity and thus has to infer unobserved world states. The Bayesian inference is a normative framework to implement inference, and extensive studies have suggested that the brain's perception is consistent with the Bayesian inference, forming the concept of Bayesian brain [1], including, e.g., visual processing [2], multisensory integration [3], decision-making [4], sensorimotor learning [5], etc. Studying how neural circuits in the brain realize Bayesian inference has been an active topic in neuroscience [6–8]. Many neural circuit models of Bayesian inference have been developed with distinct representational and algorithmic mechanisms, e.g., parametric-based representation [4, 8–11] and sampling-based representation [12–19].

Despite a large body of neural circuit models of Bayesian inference, there are still gaps between our current Bayesian circuit models and canonical recurrent circuits in the cortex. One obvious distinction is previous Bayesian circuit models haven't considered the rich diversity of neuronal types in the cortex, especially inhibitory interneurons. The canonical cortical microcircuit contains

---

*Corresponding author.

38th Conference on Neural Information Processing Systems (NeurIPS 2024).

three major types of inhibitory (I) interneurons [20–24], including Parvalbumin (PV), Somatostatin (SOM), and Vasoactive Intestinal Peptide (VIP) neurons (Fig. 1A). These interneurons have different electrical properties, stimulus-tuning profiles, distinct connectivity, and synaptic modulations with other neurons. For example, PV neurons are weakly tuned to stimulus [20], and *multiplicatively* modulate E neurons in a way called divisive normalization by sending axons to E neurons' cell body [21, 22, 25, 26]. In contrast, SOM neurons are tuned to the stimulus, and module E neurons in an *additive* way via sending axons to the distal dendrites of E neurons [22, 25, 26]. Another gap comes from the *nonlinearity* of cortical circuit dynamics, which impedes the analytical understanding of circuits' Bayesian inference algorithms. Most earlier studies relied on numerical methods to analyze the algorithm in nonlinear circuits (e.g., [15–17, 19]), or considered linear neural dynamics to obtain analytical solutions (e.g., [12, 18, 27, 28]). We still lack a comprehensive understanding about how the nonlinear recurrent circuit dynamics with diversity of interneurons implement Bayesian inference.

To gain insight into Bayesian computation in a nonlinear recurrent circuit with types of interneurons, we build a canonical recurrent circuit model consisting of excitatory (E) and two types of interneurons, including PV and SOM neurons (Fig. 1B), and investigate how the model implements sampling-based Bayesian inference. The E neurons are modeled as a rate-based ring (attractor) model that emerges the tunings over a 1D stimulus such as orientation. The E neurons receive internal Poisson-like variability mimicking stochastic spike generation, which provides a variability source to drive sampling [18]. For simplicity, the PV neurons in the model are not tuned to the stimulus as a limiting case of their weak tuning found in experiments [20], and provides global inhibition to E neurons via divisive normalization to ensure stability [21, 25]. In contrast, the SOM neurons have stimulus tunings and provide locally tuned inhibitory feedback to E neurons in an additive way [20, 29]. The circuit model with the above connectivity successfully reproduces multiplicative and additive modulations on E neurons' tunings from PV and SOM neurons respectively (Fig. 1G-H) [26].

We perform theoretical analysis on the nonlinear recurrent circuit dynamics, and analytically identify the sampling algorithm adopted in the circuit. We find the reduced circuit with only E and PV neurons can implement Langevin sampling in the stimulus feature manifold. The tuning-dependent inhibitory feedback from SOM speeds up the sampling by upgrading the Langevin sampling into Hamiltonian sampling. And the two types of interneurons have different effects on sampling speed. Moreover, we find that Hamiltonian sampling requires SOM neurons not to receive feedforward sensory inputs, consistent with neuroanatomy with few feedforward synapses targeting SOM neurons [22, 24]. The nonlinear circuit model with fixed weights can flexibly sample posteriors with different uncertainties, if located in the linear input-output regime. At last, the circuits can be extended to sample multivariate stimulus posteriors and bimodal posteriors.

## 2 The recurrent neural circuit with various types of interneurons

The cerebral cortex is a repetition of the canonical neural circuit composed of multiple types of neurons (Fig. 1A), including excitatory (E) neurons and three major types of inhibitory (I) interneurons (PV, SOM, and VIP; classified via biomarkers [30]). To study how sampling-based Bayesian inference is implemented by the canonical neural circuit composed of various types of neurons, we build a recurrent neural circuit model consisting of E neurons and two types of I interneurons of PV and SOM neurons (Fig. 1B). The model doesn't include VIP neurons, which will form our future research (see Discussion). The basic wiring diagram of the proposed circuit model is consistent with the structure of the canonical cortical circuit (Fig. 1A). In the model, the E neurons are selective for a 1D periodic stimulus feature $z \in (-\pi, \pi]$, e.g., the orientation moving direction. Denote $\theta_j$ as the preferred stimulus feature of the $j$-th E neuron, and the preferred stimulus features of all $N_E$ E neurons, $\{\theta_j\}_{j=1}^{N_E}$ uniformly cover the whole range of feature space $z$ (Fig. 1C and F). This setting is the same as the canonical ring network model that has been widely used in modeling cortical circuits (e.g.,[17, 31–36]). Mathematically, in the continuum limit ($\theta_j \to \theta$) corresponding to an infinite number of neurons, the dynamics of the E neurons is [34, 37],

$$\tau \frac{\partial \mathbf{u}_E(\theta, t)}{\partial t} = -\mathbf{u}_E(\theta, t) + \rho \sum_{X=E,F,S} (\mathbf{W}_{EX} * \mathbf{r}_X)(\theta, t) + \sqrt{\tau \mathsf{F}_E[\mathbf{u}_E(\theta, t)]_+}\xi(\theta, t), \quad (1)$$

where $\mathbf{u}_E(\theta, t)$ and $\mathbf{r}_E(\theta, t)$ are the synaptic input and firing rate respectively of the E neuron preferring $z = \theta$. $X$ denotes neuronal types with $E$, $F$ and $S$ representing E neurons, sensory feedforward inputs, and the SOM neurons respectively. $\tau$ is the time constant of synaptic input,

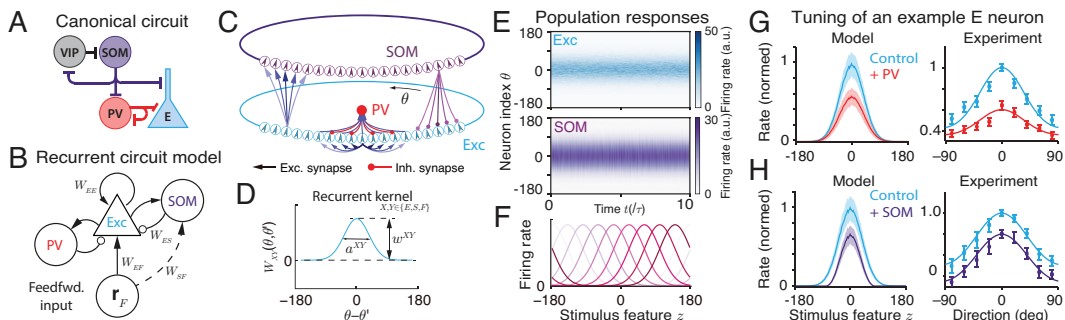

Figure 1: The recurrent circuit model. (A) The canonical cortical circuit consists of E neurons and three types of inhibitory interneurons. (B-C) The recurrent circuit model with two types of interneurons (B) and detailed recurrent circuit structure (C). (D) The Gaussian recurrent connection kernels in the circuit model. (E) An example of population responses of E (top) and SOM (bottom) neurons over time. (F) The tuning curves of E neurons in the model. (G-H) The tuning curve of an example E neuron in control state compared with enhancing PV neurons (G) and SOM neurons (H). The model result is qualitatively similar to the experimental data from [26].

and $\rho = N_E/2\pi$ is the neuronal density covering the space of stimulus feature $z$. E neurons receive an internal Poisson variability mimicking stochastic spike generation (Eq. 1, last term), and $F_E$ the Fano factor of the injected variability whose value will be adjusted to make the Fano factor of E neurons' activities at the order of 1. $\xi(\theta, t)$ is standard Gaussian white noise, i.e., $\langle \xi(\theta, t)\xi(\theta', t') \rangle = \delta(\theta - \theta')\delta(t - t')$ with $\delta(t - t')$ being the Dirac delta function. The internal Poisson variability provides the variability for the circuit to draw random samples from the posterior [18], which may arise from the E and I balance in the cortex as a complex network effect [38–40].

**Recurrent connection kernels**. $\mathbf{W}_{YX}(\theta)$ denotes the recurrent connection kernel from neurons with type $X$ to those with type $Y$, which are modeled as Gaussian functions in the model (Fig. 1D),

$$\mathbf{W}_{YX}(\theta) = w_{YX}\left(\sqrt{2\pi}a_{XY}\right)^{-1}\exp(-\theta^2/2a_{XY}^2), \qquad (2)$$

where $w_{YX}$ controls the peak strength of the recurrent weight, and $a$ the connection width across the stimulus feature space. The kernels $\mathbf{W}_{YX}(\theta)$ connecting different neuronal types can have different peak weights $w_{YX}$ ($w_{YX} > 0$ or $< 0$ regards to E or I synapses respectively). Moreover, different $\mathbf{W}_{YX}(\theta)$ may have different connection widths $a_{XY}$, although most of them have the same width $a_{XY} = a$ unless noted otherwise (see Supplementary Information (SI.) Sec. 6.1). In Eq. (1), the symbol $*$ denotes the spatial convolution, i.e., $(\mathbf{W} * \mathbf{r})(\theta) = \int \mathbf{W}(\theta - \theta')\mathbf{r}(\theta')d\theta'$, which implies the translation-invariance of the connection weight between neurons in the stimulus feature space.

**Sensory feedforward inputs**. The recurrent circuit model receives sensory feedforward input $\mathbf{r}_F(\theta, t)$ (Eq. 1) randomly evoked from a stimulus feature $z$ in the world. Given a stimulus feature $z$, we assume the feedforward inputs $\mathbf{r}_F$ are conditionally independent Poisson spikes with Gaussian tuning (Fig. 2A-B), which has been widely used before (e.g., probabilistic population code [4, 8, 9]).

$$\mathbf{r}_F(\theta|z) \sim \text{Poisson}[\lambda_F(\theta|z)], \quad \lambda_F(\theta|z) = R_F \exp[-(\theta - z)^2/2a^2], \qquad (3)$$

where $\lambda_F(\theta|z)$ is the mean firing rate. In simulating our rate-based model, the $\mathbf{r}_F$ is approximated as a continuous Gaussian random variable with multiplicative noise to mimic the Poisson statistics.

## 2.1 Inhibitory interneurons in the circuit model

**PV neurons**. The stimulus orientation weakly modulates the PV neurons [20, 26, 41], hence, for simplicity, we consider PV neurons in the model are not tuned for stimulus features and only provide global unstructured inhibition to E neurons to keep stability. Moreover, it was suggested PV neurons provide divisive normalization (DN) to modulate E neurons' responses via shunting inhibition [21, 25, 42]. Hence the proposed circuit model absorbs PV neurons' effects in the divisive normalization of E neurons which has been widely used in circuit models [21, 25, 43, 44],

$$\mathbf{r}_E(\theta, t) = \frac{[\mathbf{u}_E(\theta, t)]_+^2}{1 + \rho w_{EP}\int [\mathbf{u}_E(\theta', t)]_+^2 d\theta'}, \qquad (4)$$

where the DN acts as an activation function transferring the instantaneous synaptic input $\mathbf{u}_E(\theta, t)$ of E neurons into their firing rate $\mathbf{r}_E(\theta, t)$. $[x]_+ = \max(x, 0)$ denotes negative rectification. The integral $\int [\mathbf{u}_E(\theta, t)]_+^2 d\theta' \equiv \mathbf{r}_{PV}$, reflects PV neurons globally summing all E neurons' activities. $w_{EP}$ is the global inhibition strength characterizing the inhibitory weight from PV to E neurons.

**SOM neurons**. It is suggested that SOM neurons linearly modulate E neurons' responses, in contrast to multiplicative modulation from PV to E neurons [26]. Therefore the model considers the E neurons receive additive synaptic inputs from SOM neurons $((\mathbf{W}_{ES} * \mathbf{r}_S)(\theta, t)$, Eq. 1). Furthermore, SOM neurons are tuned to a stimulus feature with a strength comparable to E neurons [20], unlike the weak tunings of PV neurons. Thus, the dynamics of SOM neurons are governed by,

$$\tau \frac{\partial \mathbf{u}_S(x, t)}{\partial t} = -\mathbf{u}_S(x, t) + \rho \sum_{X=E,F} (\mathbf{W}_{SX} * \mathbf{r}_X)(\theta, t); \quad \mathbf{r}_S(\theta, t) = g_S \cdot [\mathbf{u}_S(x, t)]_+, \quad (5)$$

where the $g_S$ (scalar) controls the "gain" of SOM neurons and is set as a fixed value across the study. Two features about SOM neurons' connectivity are worth noting (Eq. 5). First, the model doesn't include recurrent inhibitory connections between SOM neurons, due to few mutual inhibitions between SOM neurons [22, 24]. This simplification will only change the effective gain $g_s$ of SOM neurons without affecting our conclusions of the circuit algorithm. Second, the proposed circuit model allows the existence of the feedforward connections to SOM neurons ($\mathbf{W}_{SF}$ in Eq. 5), even if they are rare in reality [22, 24]. The reason for allowing this rare connection is we want to test whether the Bayesian sampling theory has the power to constrain it.

Overall, the proposed circuit model is consistent with most canonical recurrent circuit models in the field (e.g.,[31–36]). The simplifications considered above reserve the main characteristics of neuronal connectivity and response properties observed in experiments, especially interactions between E neurons and interneurons. For example, it was found enhancing PV neurons will *multiplicatively* modulate E neurons' tuning curves (Fig. 1G, right), whereas SOM neurons modulate E neurons' tuning *additively* (Fig. 1H, right) [26]. Both effects are successfully reproduced in the proposed circuit model (Fig. 1G-H, left; see details in SI. Sec. 6.5).

## 3 From recurrent circuit dynamics to Bayesian inference

The proposed recurrent circuit dynamics (Eqs. 1 and 5) is supposed to implement Bayesian inference by computing the stimulus posterior based on a received feedforward input,

$$p(z|\mathbf{r}_F) \propto p(\mathbf{r}_F|z)p(z). \quad (6)$$

It regards the stage from external stimulus $z$ to the feedforward input $\mathbf{r}_F$ as the probabilistic generative process (Fig. 2A). Implementing Bayesian inference requires the recurrent circuit to store the generative model. Rather than defining a generative model and deriving its neural circuit implementation as in many previous studies, here we ask the question the other way around: if the proposed recurrent circuit model based on neurophysiology could do Bayesian inference (Eq. 1A), what generative model is stored in the circuit and what stimulus posteriors are computed by the circuit? Furthermore, what is the Bayesian inference algorithm adopted by the circuit dynamics?

**Stimulus likelihood**. The stochastic feedforward input from the stimulus feature $z$ (Eq. 3) naturally specifies the stimulus likelihood that can be calculated as a Gaussian likelihood (see SI. Sec. 2.1),

$$p(\mathbf{r}_F|z) = \prod_\theta \text{Poisson}[\lambda_F(\theta|z)] \propto \mathcal{N}(z|\mu_z, \Lambda^{-1}), \quad (7)$$

where the Gaussian stimulus likelihood function comes from the the Gaussian profile of feedforward input tuning $\lambda_F(\theta|z)$ (Eq. 3) [9]. The mean $\mu_z$ and the precision $\Lambda$ of the stimulus likelihood can be read out from $\mathbf{r}_F$ via a linear decoder called population vector [45, 46],

$$\mu_z = \sum_j \mathbf{r}_F(\theta_j)\theta_j / \sum_j \mathbf{r}_F(\theta_j), \quad \Lambda = a^{-2} \sum_j \mathbf{r}_F(\theta_j). \quad (8)$$

Geometrically, $\mu_z$ is regarded as the location of $\mathbf{r}_F$ in the stimulus feature space, and $\Lambda$ is proportional to the input spike count (Fig. 2B-C). In this way, a single snapshot of $\mathbf{r}_F$ parametrically conveys the whole stimulus likelihood function $p(\mathbf{r}_F|z)$ [9].

**Subjective prior**. We suppose the recurrent circuit utilizes its stored *subjective* prior to compute the subjective stimulus posteriors, with a mild assumption that the subjective prior in the circuit matches the *objective* prior in the outside world. Nevertheless, the subjective circuit prior remains unknown at this point. Next, we theoretically analyze the circuit dynamics to identify the stored subjective prior in the circuit and find out the circuit algorithm of Bayesian sampling.

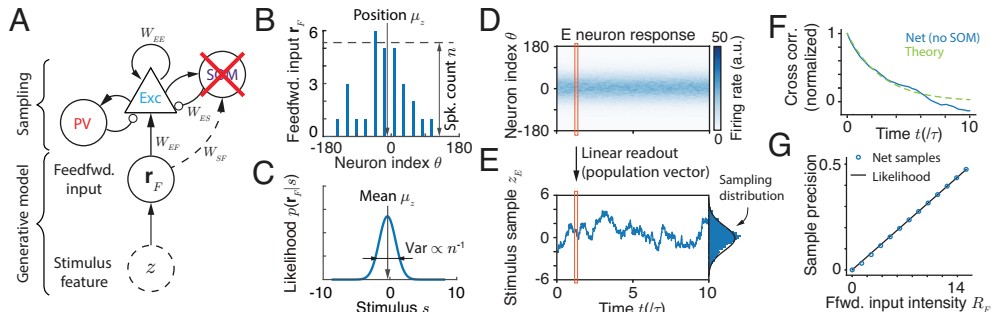

Figure 2: (A) A reduced circuit after blocking SOM neurons. (B-C) A schematic of the feedforward (spiking) input (B) and a linear read out of the stimulus likelihood conveyed by the feedforward input (C). Geometrically, the position and spike count of feedforward input determine the mean and variance of likelihood respectively. (D-E) E neurons' population responses (D) and the location of instantaneous E population response on the $y$-axis is regarded as stimulus sample $z_E$ generated by the network (E) and can be read out via a linear decoder called population factor. (F) The cross-correlation function of stimulus samples generated by the circuit. (G) The circuit with fixed parameters flexibly samples posteriors with various uncertainties (controlled by feedforward input intensity).

## 4 The Bayesian sampling in the stochastic circuit dynamics

### 4.1 Theoretical analysis of the neural dynamics

We theoretically analyze the nonlinear circuit dynamics to investigate how it can implement Bayesian sampling. We perform perturbative analysis of the nonlinear circuit dynamics, identify the low-dimensional stimulus feature manifold (subspace) in the high-dimensional population response space, and eventually study the circuit dynamics on the stimulus feature manifold. Given a feedforward input $\mathbf{r}_F$ (Eq. 3), it can be checked that the synaptic input $\mathbf{u}_X(\theta)$ and the firing rate $\mathbf{r}_X(\theta)$ of neurons $X$ in the equilibrium attractor states are both Gaussian profiles (Fig. 1E-F; see SI. Sec. 3.1) ,

$$\langle \mathbf{u}_X(\theta) \rangle = U_X \exp[-(\theta - \bar{z}_X)^2/4a^2], \quad \langle \mathbf{r}_X(\theta) \rangle = R_X \exp[-(\theta - \bar{z}_X)^2/2a^2], \quad (X = E, S) \quad (9)$$

where $\langle \cdot \rangle$ denotes the average over different realizations. $U_X$ and $R_X$ denote the height of the population synaptic input and firing rate respectively, and can be analytically computed (Eq. S37). The position of population activity on the stimulus feature space is $\bar{z}_X = \mu_z$ is the same as the location $\mu_z$ of the feedforward inputs (SI. Sec. 3.1). Intuitively, this is because the recurrent circuit is homogeneous along the stimulus feature space, i.e., all neurons are uniformly distributed on the stimulus feature space and the recurrent connections are translational invariance.

With the corruption of sensory noises and the internal Poisson variability, the instantaneous neural responses will deviate from the equilibrium attractor state (Eq. 9). We treat each instantaneous response as a perturbation from its equilibrium state, i.e., $\mathbf{u}_X(\theta, t) = \langle \mathbf{u}_X(\theta) \rangle + \delta\mathbf{u}_X(\theta, t), (X = E, S)$, and the relaxation dynamics of the perturbation $\delta\mathbf{u}_X(\theta, t)$ can be derived [47]. Then performing eigen-analysis of the perturbation dynamics we analytically find out the stimulus feature manifold (subspace), which is specified by its (unnormalized) eigenvector [35, 47, 48],

$$\phi(\theta|z_X) \propto (\theta - z_X) \exp[-(\theta - z_X)^2/4a^2], \quad (X = E, S). \quad (10)$$

Previous studies have shown the stimulus feature eigenvector has the largest eigenvalue in the perturbation dynamics [47]. Therefore, we project the dynamics of E and SOM neurons (Eqs. 1 and 5) onto their respective stimulus feature eigenvectors, where the projection is computing the inner product between the neuronal responses and the eigenvector, i.e., $\langle \phi(\theta), f(\theta) \rangle = \int \phi(\theta) f(\theta) d\theta$, with $f(\theta)$ representing the left or right handed side of Eqs. (1 and 5). The projection yields the dynamics of the E and SOM neurons on the stimulus feature manifold (see details in SI. Sec. 3.3)

$$\tau_E \dot{z}_E \approx g_{ES}(z_S - z_E) + g_{EF}(\mu_z - z_E) + \sigma_E \sqrt{\tau_E} \xi_t, \quad (11)$$
$$\tau_S \dot{z}_S \approx g_{SE}(z_E - z_S) + g_{SF}(\mu_z - z_S) \quad (12)$$

The approximation comes from ignorance of some negligible nonlinear terms. $z_E$ and $z_S$ denote the instantaneous positions of neural activities at the stimulus feature manifold at time $t$ (Fig. 2E). $\mu_z$ is the observed stimulus feature conveyed by the feedforward input (Eq. 8). $\tau_X = \tau U_X (X = E, S)$

is the time constant of the circuit dynamics on the stimulus feature manifold, where $U_X$ is the peak value of population synaptic input (Eq. 9). The coefficients $g_{XY}$ in the above equation denote the coupling strength between neural response positions, where $g_{XY} \propto w_{XY}R_Y$ with $R_Y$ the peak firing rate of the pre-synaptic neural population (Eq. S48). The $\sigma_E^2 = 8a\mathsf{F}_E/(3\sqrt{3\pi})$ is the variance of the internal variability on the stimulus feature manifold coming from the internal Poisson variability (Eq. 1, last term). $\sigma_E$ is a constant value irrelevant with feedforward inputs and network responses.

## 4.2 Langevin sampling in the reduced circuit (E and PV neurons)

Since the circuit contains multiple types of neurons, we first analyze a reduced circuit dynamics on the stimulus feature manifold without SOM neuron (Fig. 2A, setting $g_{ES}$ to zero in Eq. 11) and then study how the SOM neurons affect the sampling dynamics. Without SOM neurons (only E and PV neurons), the E neurons' dynamics on the stimulus feature manifold is,

$$\dot{z}_E = \tau_E^{-1} g_{EF}(\mu_z - z_E) + \sigma_E \tau_E^{-1/2} \xi_t, \tag{13}$$

which is a first-order Langevin dynamics. This motivates the possibility that the E and PV neurons can implement Langevin sampling in the stimulus feature manifold. If true, the instantaneous position of E activity, $z_E$ (Fig. 2D), can be regarded as a stimulus feature sample generated by the circuit. In theory, the Langevin sampling of a posterior $p(z|\mathbf{r}_F) \propto p(\mathbf{r}_F|z)p(z)$ corresponds to performing stochastic ascent on the log posterior surface [49, 50],

$$\dot{z} = \tau_z^{-1} \nabla[\ln p(\mathbf{r}_F|z) + \ln p(z)] + (\tau_z/2)^{-1/2}\xi_t, \quad (\nabla \equiv d/dz)$$
$$= \tau_z^{-1}[\Lambda(\mu_z - z) + \nabla \ln p(z)] + (\tau_z/2)^{-1/2}\xi_t, \tag{14}$$

where $\tau_z$ is the sampling time constant controlling the sampling speed. The 2nd row is obtained by substituting the Gaussian likelihood (Eq. 7).

**Uniform subjective prior**. Comparing Eqs. (14) and (13), we can identify a stored *uniform* stimulus prior $p(z)$ in the reduced circuit model stores . This is because the gradient of a uniform prior is $\nabla \ln p(z) = 0$, and then the Langevin sampling dynamics (Eq. 14) reduces to a form similar to the circuit dynamics on the stimulus manifold (Eq. 13). The uniform stimulus prior comes from homogeneous neurons in the circuit, i.e., neurons are uniformly distributed along the stimulus feature space, and the translation-invariant connection profile (Eq. 2). It implies that the circuit needs to break the symmetry of homogeneous neurons to store a non-uniform prior (see Discussion).

**Condition for realizing Langevin sampling**. Utilizing Langevin dynamics to sample the posterior requires the drift and diffusion coefficients to share the same time constant $\tau_z$ (Eq. 14). To satisfy this requirement, the E dynamics on the stimulus feature manifold (Eq. 13) should have $g_{EF}/\sigma_E^2 = \Lambda/2$. With the expressions of $g_{EF}$ and $\sigma_E^2$ (Eq. S48), the feedforward connection weight $w_{EF}$ should be,

$$w_{EF} = \frac{\sqrt{\pi}}{a}\sigma_E^2 = \left(\frac{2}{\sqrt{3}}\right)^3 \mathsf{F}_E. \tag{15}$$

Intuitively, larger internal variability $\mathsf{F}_E$ increases the sampling variance, and it requires a larger feedforward weigh $w_{EF}$ to compensate for sampling variance increase to match with the posterior variance. To verify our theoretical derivation (Eq. 15), we search whether there is an optimal value of the feedforward weight $w_{EF}$ allowing the circuit without SOM neurons to sample the posterior (likelihood). Indeed, we find once the recurrent circuit model is set with that optimal $w_{EF}$, the reduced circuit model (consist of E and PV neurons) with all parameters fixed can flexibly sample the likelihood with various uncertainties (Fig. 2G). A characteristic of Langevin sampling is the cross-correlation function of samples (Eq. 14) exponentially decays with time which can be calculated as $\rho(\Delta t) = \exp(-g_{EF}\Delta t/\tau_E)$ (SI. Eqs. S7). To verify whether the reduced circuit performs sampling with the Langevin dynamics as suggested by Eq. (13), we estimate the cross-correlation function of stimulus sample $z_E$ generated by the network, which indeed exhibits an exponential form, and our theoretical calculation $\rho(\Delta t)$ predicts the actual cross-correlation function well (Fig. 2F).

## 4.3 SOM neurons accelerate Bayesian sampling in E neurons

The SOM neurons augment the dimensionality of the circuit dynamics on the stimulus feature manifold from the first order to the second order (Eqs. 11 and 12). In principle, the second-order

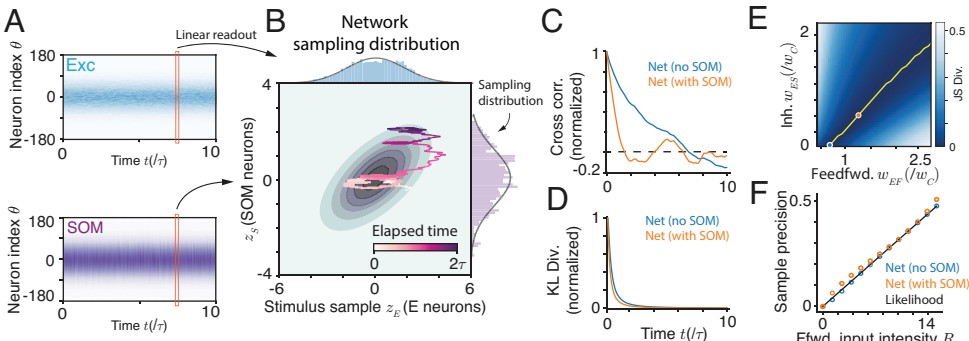

Figure 3: The Bayesian sampling in the full circuit model with PV and SOM neurons. (A) The population responses of E and SOM neurons. (B) The network's sampling distribution read out from E and SOM neurons in a way similar with Fig. 2D-E. The E neuron's position is regarded as stimulus feature sample $z_E$, while the sample of SOM neurons $z_S$ contribute to the auxiliary variable in Hamiltonian sampling. The distribution of $z_E$ (right marginal) will be used to approximate the posterior. (C-D) The cross-correlation function of stimulus sample $z_E$ (C) and the decaying of KL divergence from posterior and the sampling distribution over time (D). (D) A linear manifold as the combination of feedforward weight $w_{EF}$ and the inhibitory weight from SOM to E neurons $w_{ES}$ allowing the full circuit to sample the posterior correctly. (E) The circuit with fixed weights (Fig. 3D, orange dot) can flexibly sample posteriors with various uncertainties.

Hamiltonian sampling dynamics motivates us to explore whether and how the circuit can implement Hamiltonian-like sampling [50, 51]. The computational benefit from the increased complexity of Hamiltonian sampling compared with Langevin sampling is accelerating the sampling speed by generating less correlated samples over time [50]. To investigate such a possibility, we write one commonly used Hamiltonian sampling dynamics [50, 52, 53],

$$\tau_z \dot{z} = y; \quad \tau_y \dot{y} = -\beta y + \nabla \ln p(z|\mathbf{r}_F) + (2\beta\tau_z)^{1/2}\eta_t. \tag{16}$$

$y$ is the auxiliary variable representing momentum in the Hamiltonian sampling. It can speed up sampling by increasing sampling step size when close to the posterior center, helping the sampling trajectories move toward the other side of the posterior (Fig. 3B; details in SI. Sec. 2.1). $\beta$ is the friction strength determining how fast the momentum will decay to zero. We next bridge the circuit dynamics (Eqs. 11 and 12) with the Hamiltonian sampling dynamics (Eq. 16).

Intuitively, the sample from SOM neurons $z_S$ (Eq. 12) resemble the auxiliary variable $y$ in the Hamiltonian dynamics (Eq. 16). Nevertheless, there are several gaps between the two: First, in the Hamiltonian sampling, the stimulus sample $z$ is purely driven by the auxiliary variable $y$ (Eq. 16), whereas in the circuit model $z_E$ receives both $z_S$ and $\mu_z$ (Eq. 11). Second, Hamiltonian sampling injects variability into the auxiliary dynamics $y$ (Eq. 16), while the variability (from stochastic spike generation) is injected into the dynamics of $z_E$. To bridge the gap between the circuit model and sampling dynamics, we assume that the circuit is conducting a mixture of Langevin (Eq. 14) and Hamiltonian sampling (Eq. 16), and thus we split the E neurons' sampling dynamics into two parts,

$$\tau_E \dot{z}_E = \underbrace{[g_{ES}(z_S - z_E) + (1-\alpha_L)g_{EF}(\mu_z - z_E)]}_{y_S} + [\alpha_L g_{EF}(\mu_z - z_E) + \sigma_E\sqrt{\tau_E}\xi_t], \tag{17}$$
$$= \qquad\qquad y_S \qquad\qquad + [\alpha_L g_{EF}(\mu_z - z_E) + \sigma_E\sqrt{\tau_E}\xi_t],$$

where $\alpha_L \in [0, 1]$ denotes the proportion of feedforward input contributed by Langevin sampling. We see the $y_S$ resembles the auxiliary variable $y$ in Hamiltonian sampling (Eq. 16), which implies the auxiliary variable in the circuit is a mixture of samples from E neurons $z_E$ and SOM neurons $z_S$. Given the definition of $y_S$, the stochastic dynamics of $y_S$ can be derived as a form similar to the one in the Hamiltonian sampling (Eq. 16; details in SI. Sec. 4.1).

$$\dot{y}_S = -\beta_y y_S + \beta_E(\mu_z - z_E) + \beta_S(\mu_z - z_S) + \sigma_y \eta_t. \tag{18}$$

$\beta_y$, $\beta_E$, $\beta_S$ and $\sigma_y$ are functions of the coefficients in Eq. (17) (detailed expressions at Eq. S53). It can be checked that equilibrium distribution of the mixed dynamics is the posterior (details of Fokker-Planck approach in SI. Sec. 2.5).

**Conditions for realizing mixed sampling**. To utilize the full circuit model with SOM neurons (Eqs. 17-18) to implement Bayesian sampling, the coefficients in the circuit model should satisfy the

relations of coefficients required in the Langevin sampling (Eq. 14) and the Hamiltonian sampling (Eq. 16). First, setting up the Langevin sampling part in the circuit model (Eq. 17, blue) requires $\alpha_L g_{EF}/\sigma_E^2 = \Lambda/2$, which finally leads to,

$$w_{EF} = \sqrt{\pi}\sigma_E^2/(a\alpha_L), \tag{19}$$

whose value is $1/\alpha_L$ times the feedforward weight in the Bayesian sampling circuit without SOM neurons (Eq. 15). Second, realizing the Hamiltonian sampling part in the circuit model (Eq. 17, orange; and Eq. 18) yields three conditions shown below,

$$(a).\ \tau_z = \tau_E; \quad (b).\ \beta_S = 0; \quad (c).\ \frac{\tau_y}{1} = \frac{\beta}{\beta_y} = \frac{\Lambda}{\beta_E} = \frac{(2\beta\tau_z)^{1/2}}{\sigma_y}. \tag{20}$$

The conditions in Eqs. (19) and (20) combined enable the full circuit model with SOM neurons to implement Bayesian sampling. An important insight from Eq. (20b) is that the SOM neurons should not receive feedforward inputs directly ($w_{SF} = 0$ in Eq. 5; Fig. 1A, dashed line removed). This theoretical result is consistent with the anatomy that SOM neurons receive much fewer feedforward synapses than other types of neurons [23, 24]. In addition, substituting the detailed expression of coefficients into Eq. (20), we find there is a low-dimensional manifold in the circuit model's connection weight space for the circuit to sample the posterior $p(z|\mathbf{r}_F)$ (SI. Sec. 4.2),

$$\left(U_E^{-1}R_S\right) \cdot w_{ES} - \left[(1-\alpha_L)U_E^{-1}R_F\right] \cdot w_{EF} = \left[G(\alpha_L)U_S^{-1}R_E\right] \cdot w_{SE}. \tag{21}$$

$G(\alpha_L)$ is a nonlinear function of $\alpha_L$ which specifies the proportion of Langevin sampling in the circuit dynamics, which remains invariant with feedforward input and network activities (SI. Eq. S65). $U_X$ and $R_X$ are the height of the population synaptic input and firing rate of neuronal populations $X$ (Eq. 9). Eq. (21) implies that the sampling in the full circuit model is robust, without the need to fine-tune recurrent weights. To verify the theoretical result (Eq. 21), we fix the weight from E to SOM neurons, $w_{SE}$, and search whether there is a line manifold in the two-dimensional parameter space of $w_{ES}$ and $w_{EF}$ for the circuit correctly sample the stimulus posterior. Indeed, Fig. 3E numerically confirms the line manifold of weights under which the sampling distribution matches the posterior. Moreover, the introduction of SOM interneruons makes the cross-correlation of sample $z_E$ decay faster (Fig. 3C), suggesting speeding up sampling (Fig. 3D).

**Flexible sampling posteriors in the linear regime**. Moreover, the circuit model with *fixed weights* should be able to sample the posteriors with different uncertainties. In the circuit model, the posterior uncertainty is determined by the feedforward input rate $R_F$ (Eq. 3), where a larger input rate leading to smaller posterior uncertainty (Eq. 8). We find that when the nonlinear circuit model is located at the *linear regime*, it can flexibly sample posteriors with different uncertainties. To see this effect, we can change the feedforward input rate by multiplying a gain factor $g$, i.e., $R_F \mapsto gR_F$ that change the likelihood precision $\Lambda \mapsto g\Lambda$ (Eq. 8 and S5). If the circuit is at the linear regime, the peak value of synaptic input, $U_X \mapsto gU_X$, and the population firing rate, $R_X \mapsto gR_X$, are both multiplied with the gain factor $g$ being applied to the feedforward input. And then it can be checked Eq. (21) is still satisfied. This theoretical result is confirmed by numerical simulation (Fig. 3F) which shows the precision of circuit's stimulus samples increases with feedforward rate $R_F$ and aligns well with the likelihood. Here we adjust the recurrent E-to-E weight $w_{EE}$ to set the circuit has an approximately linear response at the range of the feedforward input rate. Fig. S1 shows if the network deviates from the linear regime, the circuit's sampling distribution will deviate from the likelihood.

## 4.4 The Bayesian sampling performance from interneurons

We further investigate how quantitative measures of sampling, e.g., sampling speed and temporal correlation of samples, will be affected by interneurons such as the inhibitory feedback weight. In principle, both sampling speed and temporal correlation can be revealed by the eigenvalues of the circuit sampling dynamics (Eqs. 11-12). When organizing the circuit sampling dynamics into a matrix form (Eqs. 11-12), $\dot{\mathbf{z}} = -\mathbf{M}\mathbf{z} + \boldsymbol{\mu}$, with $\mathbf{z} = (z_E, z_S)^\top$ and $\boldsymbol{\mu}$ lumping terms exclusive $z_E$ or $z_S$ in Eqs. (11 - 12), the eigenvalues of the circuit sampling dynamics are (SI. Eq. S69),

$$\lambda_\pm = \text{tr}(\mathbf{M}) \pm \sqrt{\text{tr}(\mathbf{M})^2 - 4\det(\mathbf{M})} \triangleq \text{tr}(\mathbf{M}) \pm \sqrt{\Delta},$$
$$\text{where} \quad \text{tr}(\mathbf{M}) = \tau_E^{-1}(g_{ES} + g_{EF}) + \tau_S^{-1}g_{SE}, \quad \det(\mathbf{M}) = \tau_E^{-1}\tau_S^{-1}g_{EF}g_{SE}. \tag{22}$$

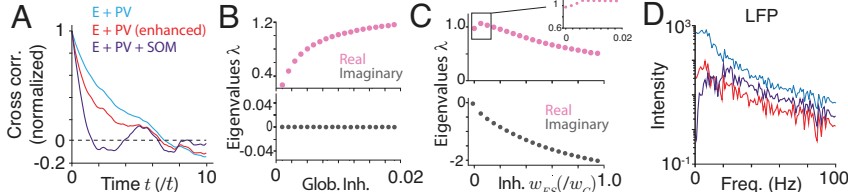

Figure 4: Interneurons' effects on sampling. (A) The cross-correlation of samples for perturbing PV and SOM. (B-C) The smallest eigenvalue of sampling dynamics when changing PV (A) and SOM (B) inhibitions. (A): obtained from the circuit without SOM (Fig. 2A). (D) The local field potential (LFP), defined by the sum of synaptic inputs of E and SOM neurons (color: same as A).

The sampling speed is limited by the smallest real part of eigenvalues, i.e., $\text{Re}(\lambda_-)$ (Eq. 22), in that the KL divergence from the distribution of samples up to time $t$ to the stationary distribution (Fig. 3D) decays exponentially as $\exp[-\text{Re}(\lambda_-)t]$ [54].

**PV neurons**. PV's inhibition weight $w_{EP}$ modulates the eigenvalues by decreasing the common factor $\tau_E = \tau U_E(w_{EP})$ (Eq. 22), where $U_E(w_{EP})$, the peak value of E population synaptic input (Eq. 9), decreases with $w_{EP}$. Hence, stronger PV inhibition increases the slowest eigenvalue $\lambda_-$ (Fig. 4A) and leads to faster sampling, exhibited by the faster decay of the temporal correlation between samples (Fig. 4A, blue and red). Moreover, the multiplicative modulation from PV neurons will not induce the imaginary part of eigenvalues (Fig. 4B, bottom), i.e., no oscillation between samples.

**SOM neurons**. The SOM inhibitory weight $w_{ES}$ will have non-monotonic effects on sampling speed measured by the real part of the slowest eigenvalue. There is a value of $w_{ES}$ to maximize the sampling speed ($\text{Re}(\lambda_-)$ (Fig. 4C, top). Moreover, SOM's inhibition will induce temporal oscillations between samples, i.e., emerging imaginary part of eigenvalue $\lambda_-$ (Fig. 4C, bottom). This oscillation is confirmed by the cross-correlation of samples (Fig. 4A, purple). Moreover, to mimic neural experimental data analysis, the oscillation induced by SOM neurons can be revealed by the power spectrum analysis of the local field potential (LFP) (SI. Eq. S90).

## 5 Sampling complex posteriors in canonical recurrent circuits

**High-dimensional stimulus posteriors**. As a proof of concept example, we consider sampling bivariate stimulus posteriors by coupled circuits (Fig. 5A) with each circuit the same as Fig. 1B. And only E neurons across circuits are coupled. Each circuit $m$ ($m = 1, 2$) receives a feedforward input generated by a latent stimulus feature $z_m$, and will sample $z_m$. Hence, the number of coupled circuits equals to the stimulus feature dimension. We found the coupled circuits store an associative (subjective) prior, i.e., $p(z_1, z_2) \propto \exp[-\Lambda_s(z_1 - z_2)^2/2]$ (Fig. 5C), with the prior precision $\Lambda_s$ increasing with inter-circuit coupling weights (Fig. 5D; Eq. S85). Math analysis is presented in SI. Sec. 5. Concatenating the samples generated by E neurons in two circuits (with the same readout as Fig. 3A-B), we can obtain the 2D posterior sampled from the coupled circuit (Fig. 5B). Similarly, the SOM interneurons speed up sampling, reflected by the sampling trajectories with SOM transverse over a wider posterior region than the one without SOM in the same period (Fig. 5B).

**Bimodal stimulus distributions**. We also use the circuit (Fig. 1C) to sample uni-variate bimodal posteriors (Fig. S2), in response to superpositions of two feedforward inputs with each generated by a latent stimulus. The circuit can sample a bimodal distribution, where samples jump between two modes alternatively, due to the bi-stability in the circuit dynamics (Fig. S2C and E). In contrast, without SOM neurons (Fig. 2A), the circuit can only sample a uni-modal distribution, as a uni-modal approximation of the bimodal one (Fig. S2F). Due to the space limit, we haven't comprehensively linked the circuit' bimodal sampling distribution with posteriors, which will form our future work.

## 6 Conclusion and Discussion

The present study investigates how canonical recurrent circuits with diverse inhibitory interneurons implement Bayesian sampling. The nonlinear circuit model consists of E neurons, and two types of interneurons neurons (PV and SOM neurons). PV and SOM neurons have distinct tuning properties and modulations on E neurons: PV neurons have weak stimulus tuning and multiplicatively modulate E neurons, while SOM neurons have stimulus tuning and send additive inputs to E neurons. Through

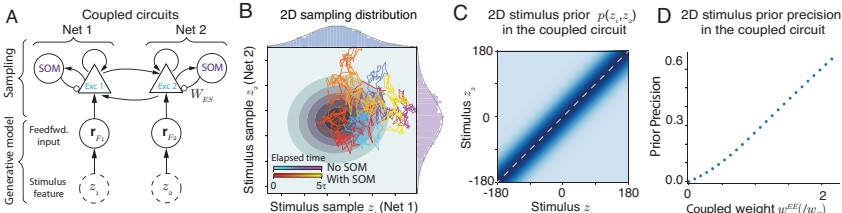

Figure 5: Sampling 2D posteriors in coupled circuits. (A) Each circuit (net) is the same as the one in Fig. 1B (PV neurons are not shown). (B) The 2D sampling distribution generated by the coupled circuit shown in (A). (C-D) The coupled circuits store an associative (subjective) prior of two stimuli (C). The prior correlation increases with inter-circuit coupling weights (D).

theoretical analysis and numerical simulations, we find the reduced circuit with only E and PV neurons implements Langevin sampling in the stimulus manifold (subspace) to compute the stimulus posterior. The SOM neurons accelerate the circuit sampling speed by upgrading the Langevin sampling into Hamiltonian sampling. We further find Hamiltonian sampling requires the SOM neurons to receive no feedforward connections, consistent with neuroanatomy. We also investigate how inhibitory strength from two types of interneurons affects the sampling speed. Our work is one of the earliest studies investigating the Bayesian sampling algorithm in canonical nonlinear recurrent circuits with diverse interneurons, and provides new insight into interneurons in Bayesian computation.

**Comparison with other work**. The computational mechanism of accelerated sampling by SOM neurons is similar to previous studies considering structured inhibitory feedback to E neurons (e.g., [14, 17, 53, 55]), while some notable differences exist. Earlier studies considered the sampling in neural response space where the posterior dimension is the same as the number of neurons [14, 17], whereas the current circuit samples in the stimulus feature manifold (subspace) in the neural response space. On the other hand, although other previous studies considered sampling in the stimulus feature manifold similar to the current model [53, 55], their structured inhibitory feedback comes from the biophysical mechanism within single neurons, e.g., spike frequency adaptation [53], and potassium channels [55]. From a neurobiology perspective, SOM neurons can be modulated by VIP neurons (Fig. 1A), whereas intracellular mechanisms are hardly modulated [53, 55], suggesting accelerated sampling from SOM neurons might be more flexible than single neuron mechanism in reality. Moreover, the proposed circuit model is similar to a recent one [18] in terms of utilizing internal Possion variability to draw samples. However, the current circuit model is nonlinear while the other study considered a linear circuit model [18]. Lastly, the coupled circuit sampling of bivariate posteriors was also considered in [34], but which didn't figure out the circuit's sampling algorithms.

**Limitations and extensions of the model**. The proposed circuit model doesn't include VIP neurons that exclusively target SOM neurons (Fig. 1A). Our future work will incorporate them and study their effects on circuit sampling. Based on our current conclusion, it is likely that VIP neurons act as a "knob" to modulate the sampling speed, depending on task needs, by changing the activation level ("gain") of SOM neurons. Moreover, we find the proposed canonical circuit stores a uniform stimulus prior, due to the homogeneity of neurons distributing on the stimulus manifold. The homogeneous neuron simplification has been widely used in neural coding and continuous attractor networks (e.g., [31, 34, 35, 56]), which simplifies the math analysis without altering results substantially. Nevertheless, the circuit has to break the neuronal homogeneity to store a non-uniform prior [57], and certainly, cortical neurons are heterogeneous. The neuronal heterogeneity can be realized by manipulating the translation-invariant recurrent connection matrix (Eq. 2), e.g., introducing randomness (zero mean with certain variance) on recurrent weight which has also been widely used in (chaotic) Excitation and Inhibition (E/I) balanced networks ([38–40, 58]). A potential function of heterogeneity from random recurrent weights is that this puts the spiking networks into the chaotic regime where the network internally generates Poisson variability, which is statistically equivalent to the injected multiplicative variability in our rate-based network (Eq. 1, last term). Moreover, the proposed circuit model only infers a static stimulus, and we will extend to infer a dynamic stimulus described by a hidden Markov model in the future. All of these form our future research.

**Acknowledgments**

W.H.Z. is supported by the UT Southwestern Endowed Scholars program. The authors thank Chengcheng Huang for fruitful comments.

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
