# OpenReview forum: "The Bayesian sampling in a canonical recurrent circuit with a diversity of inhibitory interneurons"
_NeurIPS.cc/2024/Conference — NeurIPS 2024 poster_

### Official Review · Reviewer_RifV · 2024-07-03

**Soundness:** 3
**Presentation:** 4
**Contribution:** 2
**Rating:** 7
**Confidence:** 4

**Summary:**

This paper investigates how canonical recurrent circuits in the cortex can implement sampling algorithms. The authors show that circuits with only E and PV neurons implement Langevin dynamics, while including SOM neurons enables a mixture of Langevin and Hamiltonian sampling.

**Strengths:**

The paper demonstrates its due diligence to connect canonical circuits and Bayesian sampling rigorously. Overall, the paper is well-written, and the experimental results are well-illustrated.

**Weaknesses:**

Two major limitations of the sampling circuit proposed in the paper are its low-dimensionality and uniform prior. Overall, the authors only demonstrate sampling of 1-D Gaussian distributions, due to the linear drift. While I understand that it is the norm of papers in neural sampling theory, recent works have shown that recurrent circuits are capable of sampling from much more complicated distributions [1,2]. The authors should discuss how the proposed circuits can scale in more detail.

[1] Lyo, Benjamin and Cristina Savin. “Complex priors and flexible inference in recurrent circuits with dendritic nonlinearities.” bioRxiv (2023): n. pag.
[2] Chen, Shirui, et al. "Expressive probabilistic sampling in recurrent neural networks." Advances in Neural Information Processing Systems 36 (2024).

Minor:
1. Figure 3 caption does not include panel F.
2. Although it is straightforward for the linear case (S36), for the mixture of general Langevin and Hamiltonian dynamics (S34 & S35), it is worthwhile to include references or show that the marginal distribution of z is indeed p(z) (is that true in the general case?).

**Questions:**

I am curious about how the authors would construct a distributed version of the proposed circuit that can sample from high-dimensional stimulus posterior.

**Limitations:**

See weaknesses

---

> ### Author Rebuttal · Authors · 2024-08-05
>
> We appreciate the reviewer's feedback and positive comments about the math rigor and biological plausibility of our work.
>
> > Two major limitations of the sampling circuit proposed in the paper are its low-dimensionality and uniform prior. Overall, the authors only demonstrate sampling of 1-D Gaussian distributions, due to the linear drift. While I understand that it is the norm of papers in neural sampling theory, recent works have shown that recurrent circuits are capable of sampling from much more complicated distributions [1,2]. The authors should discuss how the proposed circuits can scale in more detail.
> >
> > [1] Lyo, Benjamin and Cristina Savin. “Complex priors and flexible inference in recurrent circuits with dendritic nonlinearities.” bioRxiv (2023): n. pag. [2] Chen, Shirui, et al. "Expressive probabilistic sampling in recurrent neural networks." Advances in Neural Information Processing Systems 36 (2024).
>
> Thank you, we very much appreciate you bringing this issue.
> We will include the discussion of recent papers that sampled more complicated posteriors in the revised manuscript, and do some comparison.
>
> In addition, the proposed circuit with SOM neurons can sample multi-modal and high-dim posteriors. Please see the results in the rebuttal PDF and our response in Global Rebuttal.
> We are happy to include the results to strengthen our paper in the revised version with one more page, if you encourage us to do that.
> The reason for not including them is because of the page limit and the completeness of the manuscript, where we sacrificed the inference task complexity to present the reasoning of building the model and math analysis.
>
> Compared with the two papers mentioned by the reviewers, our study presented a simpler inference task, compensated by the analytical results of identifying sampling algorithms in the circuit, and biological plausibility (diverse interneurons, reproducing experimentally observed tuning curve). Our rebuttal PDF shows the proposed circuit model can also sample multi-modal and high-dim posteriors. We wish the reviewer could re-evaluate the significance of our model framework and analytical methodology.
>
>
> > Minor: Figure 3 caption does not include panel F.
>
> Apologies for the typo. We will add it to the revised manuscript.
>
> > Minor: Although it is straightforward for the linear case (S36), for the mixture of general Langevin and Hamiltonian dynamics (S34 & S35), it is worthwhile to include references or show that the marginal distribution of z is indeed p(z) (is that true in the general case?).
>
> The mixture of Langevin and Hamiltonian (Eqs. S34 & S35) was used by Ref. 14 and 50, and it is guaranteed to sample the correct $p(z)$.
> We will emphasize this property in the revised manuscript.
>
> > I am curious about how the authors would construct a distributed version of the proposed circuit that can sample from high-dimensional stimulus posterior.
>
> Please see our results of extending the proposed circuit model to sample high-dim posteriors in the Global Rebuttal and attached Rebuttal PDF.
>
> To sample high-dim posteriors, we can extend the current model into many coupled networks with each the same as the one in the current manuscript. Each network receives a neural input generated from a latent 1D stimulus, and samples the corresponding 1D latent stimulus. As a whole, the coupled networks distributivity sample high-dim posteriors, where the dimension of the posterior is determined by the number of networks in the coupled circuits. The recurrent weights between subnetworks store a high-dim correlation prior of stimuli. In this way, the coupled networks are similar to the network presented in Zhang et al., Nat. Comms. 2023 (see its Fig. 6), but each subnetwork in Zhang's study didn't consider SOM neurons.

---

> > ### Comment · Reviewer_RifV · 2024-08-08
> > **Follow-up question**
> >
> > Could the author be more specific about where in Ref. 14 and 50 this mixture of Langevin and Hamiltonian is used?

---

> > > ### Author Response · Authors · 2024-08-08
> > >
> > > Thanks for the reviewer's reply!
> > >
> > > In Ref. 14, Eqs. 14-17 stated mixing the Langevin and Hamiltonian sampling dynamics will not perturb the stationary distribution, although they didn't provide rigorous math proof.
> > > Unlike Ref. 14 which considered the classical Hamiltonian dynamics, the present study considered a modified Hamiltonian dynamics with friction whose mathematical form is the same as the Eq. 9 in Ref. 50. Then we mix this modified Hamiltonian dynamics with the Langevin dynamics, which yields the Eqs. S34-S35..
> > >
> > > We are sure mixing two __linear__ SDEs with the same equilibrium distribution will not change the equilibrium distribution, which can be easily proved by writing down the Fokker-Planck equation of the SDE. Nevertheless, we are not sure whether this is still held for nonlinear SDEs.

---

> > > > ### Comment · Reviewer_RifV · 2024-08-09
> > > > **Thank you**
> > > >
> > > > Thanks for the authors' further reply, the last sentence is what I was asking, I said in the original comments that the linear case is straightforward. If it only works for the linear case, that brings another problem, as $\nabla \log p(z)$ is linear iff p(z) is Gaussian, does that mean any distribution that the circuit can sample from is at best mixture of Gaussian? Natural stimulus is known for its high-order statistics, while Gaussian is completely described by its first two moments. Although I don't think one should a priori assume the capability of the neural circuits (that's the job of experimentalists), I am very skeptical that neural circuits can only sample from distributions of Gaussian nature. Why consider nonlinear neural circuits when you just need to sample from Gaussian?
> > > >
> > > > I also read the authors' rebuttal to all reviewers
> > > > > Studied a nonlinear, biologically plausible dynamics but without analytical results of the nonlinear dynamics (e.g., Benjamin, ICLR 2024; Chen, NeurIPS 2023; Echeveste, Nat. Neurosci., 2020).
> > > >
> > > > This is not true, Chen, NeurIPS 2023 and Echeveste, Nat. Neurosci., 2020 both have substantial analytical results. The diffusion model used in Benjamin, ICLR 2024 is also heavily backed by analytical results. Maybe the authors can be more specific on what their contribution is? For example, the Recurrent connection kernels proposed is also studied in e.g. Dong et al., 2022, where it is called CANN, what is the difference between CANN and the recurrent connection kernels?

---

> ### Author Response · Authors · 2024-08-09
> **Invariant equilibirum distribution with mixing Langevin and Hamiltonian**
>
> #### Invariant equilibrium distribution by mixing Langevin and Hamiltonian dynamics
>
> We performed some theoretical analysis to show the mixture of Langevin and Hamiltonian dynamics will not change the equilibrium distribution.
> We start by copying the equations of Hamiltonian dynamics with friction (Eq. S13-14) here,
> $$
> \begin{align}
> \tau_z \dot{z} = y; \\
> \tau_y \dot{y} = -\nabla U(z)  - \beta y + (2\beta \tau_z)^{1/2} \eta_t, \quad\quad (1)
> \end{align}
> $$
> where $U(z) = -\log p(z)$ is the logarithm of the distribution being sampled.
>
> To intuitively and quickly show the equilibrium distribution of $p(z)$, we consider a case where the time scale of $y$ (determined by the time constant $\tau_y/ \beta$) is much faster than the time scale of $z$, i.e., $y$ changes much faster than $z$. And then we could approximately treat the $z$ is fixed during the time scale of $y$, and then we utilize __time separation__ to analyze the two dynamics separately (this strategy was also used in the paragraph around Eqs. 11-12 in Ref. 50).
> The equilibrium distribution of $y$ can be solved as
> $$p(y) = \mathcal{N}[y|-\beta^{-1}\nabla U(z), \tau_z/\tau_y]\quad\quad (2)$$
> Next, we consider the change of $z$. At the time scale of $z$, we can approximately treat the $y$ as a sample from its equilibrium distribution, i.e., $y\sim p(y)$, and write $y$ as
> $$y = - \beta^{-1}\nabla U(z) + (\tau_z/\tau_y)^{1/2} \xi_t$$
> Substituting the above equation back to the Eq. (1) shown above,
> $$
> \begin{align}
> \dot{z} = - \tau_z^{-1}\beta^{-1} \nabla U(z) + (\tau_z \tau_y)^{-1/2} \xi_t \quad\quad (3)
> \end{align}
> $$
> As long as $\beta = 2\tau_y$, the above $z$ dynamics embedded in the Hamiltonian dynamics can be treated by performing Langevin sampling dynamics to sample the distribution $p(z) \propto \exp[-U(z)]$.
>
> Then consider another Langevin dynamics to sample $p(z) \propto \exp[-U(z)]$(e.g., Eq. S33 in the Supplementary),
> $$ \dot{z} = [-\tau_L^{-1} \nabla U(z) + (\tau_L/2)^{-1/2} \eta_t] $$
> and mix it with the above Eq. (3),
> $$ \dot{z} = [-\tau_z^{-1}\beta^{-1} \nabla U(z) + (\tau_z \tau_y)^{-1/2} \xi_t] + [-\tau_L^{-1} \nabla U(z) + (\tau_L/2)^{-1/2} \eta_t] = -[\tau_z^{-1}\beta^{-1} + \tau_L^{-1}]  \nabla U(z) + [(\tau_z \tau_y)^{-1} +   (\tau_L/2)^{-1}]^{1/2} \epsilon_t $$
>
> Denoting the drift coefficient $\tau_z^{-1}\beta^{-1} + \tau_L^{-1} \equiv \lambda $ and using the above condition $\beta = 2\tau_y$, we can find the diffusion coefficient above is $\sqrt{(\tau_z \tau_y)^{-1} +   (\tau_L/2)^{-1}} = \sqrt{2\lambda}$.
> Therefore the $z$ dynamics in the mixed Langevin and Hamiltonian dynamics is
> $$ \dot{z}_t = \lambda \nabla U(z) + \sqrt{2\lambda} \epsilon_t $$
> Its equilibrium distribution is  $p(z)\propto \exp[-U(z)]$.
> Hence the mixture of Langevin and Hamiltonian dynamics can also sample the same distributions $p(z)$.
> Since $U(z)$ is general in the above derivation, the mixed dynamics can sample other types of distributions than Gaussian.
>
> To sample non-Gaussian distributions, we need to modify the recurrent connection profile (Eq. 2). Please find our reply to Reviewer __K6Di__ (in the section "non-Gaussian stimuli").
>
> PS: sorry that our wording confused you. You are right that an SDE can only sample a Gaussian $p(z)$ if its drift term is linear with $z$, whereas the linear in our last reply meant the drift term is linear over $\nabla \log p(z)$.

---

> ### Author Response · Authors · 2024-08-09
> **Dynamical system theoretical analysis, and our contribution**
>
> Thanks for giving us a chance to elaborate on our contribution.
>
> The theoretical/analytical results we meant is __dynamical system__ analysis of the nonlinear neural dynamics, e.g., perturbative analysis and bifurcation theory, whose details are included in Supp. Sec. 3. Without these analyses, we cannot __analytically__ obtain the sampling dynamics in the low-dim stimulus manifold (Eqs. 11-12) embedded into the high-dim neural dynamics.
> We agree that Chen 23, Benjamin 23, and Echeveste 20 have elegant theoretical results, but they are not the dynamical system analyses performed in the present study. For example, our theoretical analysis corresponds to theoretically analyzing the recurrent dynamics in Eq. 8 in Chen 23 to find its attractor states, eigen-spectrum analysis, dimensionality-reduction etc.
>
> ### What do we gain from neural dynamical system analysis?
> The dynamical system analysis (Supp. Sec. 3) especially the analytical results (Eqs. 11-12) significantly gain our understanding of __fine structure__ about sampling neural dynamics. For example, our study's nonlinear circuit dynamics is similar to Eq. 8 in Chen 23 in principle, both of which are Langevin dynamics. After our dynamical system analysis, we find the sampling dynamics embedded in the stimulus manifold $z$ can vary from Langevin to Hamiltonian sampling, although the neural dynamics are still Langevin. In addition, the proposed circuit with Langevin neural dynamics has the potential to implement __Levy__ sampling in the stimulus manifold, if we increase the inhibitory feedback weight from SOM to E neurons further and also inject noises into SOM's dynamics, with a similar math mechanism as Dong, NeurIPS 2021 and Sakaguchi, 2021, J. Physical Society of Japan.
> Basically, we feel the nonlinear neural dynamics has a rich repertoire to implement various sampling algorithms by using dynamical bifurcations, i.e., the proposed neural dynamics are located at three different bifurcation regimes in realizing Langevin, Hamiltonian, and Levy sampling.
> These fine dynamical mechanisms to implement neural sampling cannot be revealed without analytical results of dynamical system analysis (the results Eqs. 11-12 derived from Eqs. 1-4) as presented in the current study.
>
> ### The contributions and differences compared with earlier studies on the _computational_ side.
>
> 1. Previous studies (Chen 23, Benjamin 23, and Echeveste 20) all considered neural dynamics directly sample in the neural response $r$ space, while the neural dynamics in the present study considers the sampling in a low-dim stimulus manifold $z$ embedded in the neural response $r$ space, similar to Dong 22.
>
> 2. The proposed circuit with __fixed weights__ can sample all posteriors $p(z|r_F)$ under different observation $r_F$ which can have different uncertainties (Fig. 2G, 3F; Eq. 21). This is realized by embedding the low-dim sampling in the high-dim neural dynamics and the multiplicative variability in neural dynamics (Eq. 1, last term), where the overall neural firing rate in a population can carry the precision of distributions, i.e., the drift coefficient $g_{XY} \propto w_{XY} R_Y$ in Eqs .(11-12) is proportional to population firing rate $R_Y$ in line 188 .
> The flexibility of sampling a family of posteriors by neural dynamics with fixed parameters is a more stringent requirement (see reasons in Global Rebuttal), while it is unknown whether Chen 23 and Benjamin 23 could achieve this. Echeveste 20 had this numerical result (its Fig. 2) but it did not perform dynamical system analysis as in the present study so its underlying dynamical system mechanism is not clear. Dong 22 considered a similar network model to ours, while we are concerned that Dong's network cannot achieve this either, because the variance of injected variability (\sigma_V in its Eq. 13) needs to be re-adjusted based on instantaneous feedforward input (the 5th line after Eq. 20 in Dong 22).

---

> ### Comment · Reviewer_RifV · 2024-08-09
>
> >Therefore, when we mix the above Eq. (3) with another Langevin sampling dynamics (e.g., Eq. S33 in the Supplementary), the mixed Langevin and Hamiltonian dynamics will still sample the same distribution... And then the dynamics can also sample distributions apart from Gaussian distributions.
>
> This is exactly what I think is sketchy here, I don't think it holds true in general. In particular, your SDE is in z and y, but $\nabla \log p(z)$ can be nonlinear in $z$. Can you elaborate?

---

> > ### Author Response · Authors · 2024-08-10
> >
> > Dear reviewer:
> >
> > I just realized the display of Markdown at OpenReview is different from my laptop and the math equation numbers disappeared in OpenReview, which may confuse you about which equation is Eq. (3) in our last reply. Hence we revised our math derivation in the last reply and we hope the current derivation addresses your question about why the $p(z)$ is unchanged. If you feel our derivation by separating time scales of $z$ and $y$ is limited, we can provide a more rigorous version.
> >
> > In summary, by using time scale separation of $z$ and $y$, the $z$ dynamics in the Hamiltonian sampling is effectively a Langevin sampling dynamics of $p(z)$. Then by mixing two Langevin dynamics sampling the same equilibrium distribution, the equilibrium distribution remains unchanged.
> >
> > Yes, in our derivation $\nabla \log p(z)$ is general and can be nonlinear of $z$.

---

> ### Comment · Reviewer_RifV · 2024-08-10
>
> >$\dot z_t = \lambda\nabla U(z) + 2\lambda \epsilon_t$ Its equilibrium distribution is $p(z)\propto \exp[U(z)]$.
>
> This is incorrect. $p(z)\propto \exp[U(z)/(2\lambda)]$. So even in your simplified version of the proof, the mixture does not sample from the correct distribution. EDIT: I realize that this must be a typo of the authors. But the following paragraph still describes my doubt.
>
> Furthermore, I am also not sure how valid the approximation is, your S34 and S35 sure do not mix the Langevin and Hamiltonian in this way. S34 and S35 is a 2D SDE, and when you write down the Fokker-Planck equation, it is not obvious at all what is the equilibrium distribution.

---

> > ### Comment · Reviewer_RifV · 2024-08-10
> >
> > I figured it out, this can be proved using the Fokker-Planck equation. It can be shown that the joint stationary distribution is $p(y,z)\propto \exp(-\frac{y^2}{2} + U(z))$. Scale separation method that the authors gives critically depends on the assumption that y is faster than z.

---

> ### Comment · Reviewer_RifV · 2024-08-10
>
> I am actually okay with it if the authors cannot provide rigorous proof. But I would be very interested if the authors know any.
>
> I also recommend the authors think more about how the recurrent weights between subnetworks store not only a high-dim correlation prior of stimuli but also other high-order statistics in order to represent high-dim complex posterior.
>
> Given the other additions that the author contributed, I am happy to recommend acceptance.

---

> > ### Author Response · Authors · 2024-08-10
> > **Thanks!**
> >
> > > I also recommend the authors think more about how the recurrent weights between subnetworks store not only a high-dim correlation prior of stimuli but also other high-order statistics in order to represent high-dim complex posterior.
> >
> > This is a very important question we are interested in exploring. Regardless of the biological plausibility, one possibility is introducing a __gating mechanism__ on the recurrent weights between subnetworks, similar to gated recurrent networks. Given a value of the gating variable, the network stores a correlation prior. Combining all values of the gating variable, the network can store a __mixture__ of correlation priors, which may emerge in high-order prior statistics. This is a possibility we would like to explore in the near future.
> >
> > > Given the other additions that the author contributed, I am happy to recommend acceptance.
> >
> > Thanks very much for your positive reply!

---

> ### Author Response · Authors · 2024-08-10
> **Calculating the equilibrium distribution via Fokker-Planck approach**
>
> Sorry for our typos. We have carefully checked our last reply and have corrected math typos, including the diffusion term in the last equation (should be $\sqrt{2\lambda} \epsilon_t$), and redefine the $U(z) = - \log p(z)$ in the Eq. (1) in the last reply by adding a minus sign to make it consistent with conventional definitions.
>
> ### Calculating the equilibrium distribution $p(z)$ in the mixed dynamics via Fokker-Planck approach
> Here we will provide a rigorous analysis showing the Eqs. (S34-S35) can correctly sample the desired distribution $p(z)$. Due to limited space, we rely on derivations used in Chen, ICML 2014 to explain the math mechanism.
> We start by defining the (equilibrium) distribution to be sampled and the corresponding Hamiltonian (Eq. S8),
> $$\pi(z,y) = \exp[-H(z,y)]; \quad H(z,y) = U(z) + K(y) = -\ln p(z) + y^2/2$$
> And the Hamiltonian sampling with friction (Eqs. S13-S14) is (removing the last two terms of the Langevin step in Eq. S34)
> $$
> \tau_z \dot{z} = y; ~~
> \tau_y \dot{y} = - \nabla U(z) - \beta y + (2\beta \tau_z)^{1/2} \eta_t
> \quad\quad (R1)
> $$
> which is analogous to the Eq. (9) in Chen, 2014.
> To emphasize the main mechanism, we consider a simple case where $\tau_y = \tau_z = 1$. For general cases of $\tau_y$ and $\tau_z$, we can set $\beta$ appropriately to sample the $p(z)$ (Eq. 20 in manuscript).
> Denoting by the vector $Z = (z,y)^\top$, we could convert the Eq. (R1) into the matrix notation,
> $$
> \frac{dZ}{dt} = -
> \begin{pmatrix} 0 & -1 \\\\ 1 & \beta \end{pmatrix}
> \begin{pmatrix} \nabla U(z) \\\\ y \end{pmatrix} + \sqrt{2}
> \begin{pmatrix} 0 & 0 \\\\ 0 & \beta^{1/2}  \end{pmatrix} {\boldsymbol \eta}_t
>  = - (D + G) \nabla H(Z) dt + (2D)^{1/2} {\boldsymbol \eta}_t
> \quad \quad (R2)
> $$
> where
> $G = \begin{pmatrix} 0 & -I \\\\ I & 0 \end{pmatrix}
> $
> is an anti-symmetric matrix,
> $D = \begin{pmatrix} 0 & 0 \\\\ 0 & \beta \end{pmatrix}
> $,
> and $\nabla H(Z) = [\partial_z H(z,y), \partial_y H(z,y)]^\top = (\nabla U(z), y)^\top$.
> Now, the Eq. (R2) is the same as the unnumbered equation in Theorem 3.2 in Chen 2014.
> Similar to the Eqs. (20) and (21) in the Supplementary in Chen 2014,
> the Fokker-Planck equation of the above dynamics is
> $$
> \partial_t p_t(Z) = \nabla^\top \{ (D+G)[\nabla H(Z)p_t(Z)]\} + \nabla^\top[D\nabla \partial_t p(Z)]
> \quad \quad (R3)
> $$
> Then utilizing the math property of the anti-symmetric $G$, we have $\nabla^\top [G \nabla p_t(Z)] = - \partial_z \partial_y p_t(z,y) + \partial_z \partial_y p_t(z,y) = 0$, then inserting this zero term into the last term of Eq. (R3),
> $$
> \nabla^\top[D\nabla \partial_t p(Z)] =  \nabla^\top[(D+G)\nabla \partial_t p(Z)]
> $$
> Substituting the above equation back to Eq. (R3), and taking the common term $(D+G)$ out of the bracket in the RHS, the Eq. (R3) can be converted into
> $$
> \partial_t p_t(Z) = \nabla^\top \{ (D+G)[\nabla H(Z)p_t(Z) +\nabla \partial_t p(Z)]
> \}
> \quad \quad (R4)
> $$
> It can be easily checked that $\pi(z,y) \propto \exp[-H(z,y)]$ is indeed the equilibrium distribution of the Eq. (R2) by calculating
> $$\nabla H(Z) \exp[-H(z,y)] +\nabla \exp[-H(z,y)] = 0$$
> Upon this point, we complete the reasoning showing the Hamiltonian dynamics with friction can sample $\pi(z) \propto \exp[-U(z)]$. And above is our restatement in explaining the math derivations underlying Theorem 3.2 in Chen 2014.
>
> Next, we mix the Hamiltonian dynamics (Eq. R2) with the Langevin dynamics of $z$ (as we did in Eqs. S34-S35), then the Eq. (R2) becomes
> $$
> \frac{dZ}{dt} = -
> \begin{pmatrix} \tau_L^{-1} & -1 \\\\ 1 & \beta \end{pmatrix}
> \begin{pmatrix} \nabla U(z) \\\\ y \end{pmatrix} + \sqrt{2}
> \begin{pmatrix} \tau_L^{-1/2} & 0 \\\\ 0 & \beta^{1/2} \end{pmatrix} {\eta}_t
>  = - (D' + G) \nabla H(Z) dt + (2D')^{1/2} \eta_t
> \quad \quad (R5)
> $$
> where the
> $
> D' = \begin{pmatrix} \tau_L^{-1} & 0 \\\\ 0 & \beta \end{pmatrix}
> $.
> Comparing Eq. (R5) with Eq. (R2), we find mixing the Langevin dynamics only changes the notation of matrix $D$ (appearing in both drift and diffusion terms) into a new matrix notation $D'$. Therefore the corresponding Fokker-Planck equation of the Eq. (R5) will be similar with Eq. (R4) but just replacing the $D$ by $D'$
> $$
> \partial_t p_t(Z) = \nabla^\top \{ (D'+G)[\nabla H(Z)p_t(Z) +\nabla \partial_t p(Z)]
> \}
> $$
> And then the equilibrium distribution of the mixed dynamics (Eq. R5) is also $\pi(z,y)$, i.e., mixing the Langevin dynamics with Hamiltonian dynamics (Eqs. S34-35) doesn't change the equilibrium distribution $p(z)$.
>
> We plan to incorporate a more detailed version of the above math derivation in a new Section 2.4 in our Supplementary to strengthen our theoretical analysis.

---

### Official Review · Reviewer_K6Di · 2024-07-11

**Soundness:** 3
**Presentation:** 2
**Contribution:** 2
**Rating:** 5
**Confidence:** 3

**Summary:**

The paper is build on previous work on Bayesian sampling to provide incremental but novel analytical insights about a neuronal circuit implementing bayesian inference. The circuit comprises excitatory neurons and two types of inhibitory neurons, PV and SOM, and uses the ring architecture. Authors find that the dynamics of such a circuit performs a Bayesian sampling algorithm and analyse analytically the properties of such sampling. They find that the circuit with E neurons and PV interneurons performs Langevin sampling while including SOM interneurons, the circuit performs Hamiltonian sampling.

**Strengths:**

The paper is build on previous work on bayesian sampling to provide incremental but novel analytical insights about a neuronal circuit implementing bayesian inference. The topic of Bayesian inference is in general of interest to the NeurIPS community and the paper builds on latest results in the field. While I have not checked the Equations in detail, the paper seems technically sound.

**Weaknesses:**

The biggest weakness of the paper is its convoluted style of presentation - the paper is hard to read. Authors could dedicate more space to give intuitive explanations about the meaning of the main results. For example, authors could give some intuition about Langevin and Hamiltonian sampling and why having different types of interneurons gives different types of sampling. The meaning of the main results of the paper remains largely unclear.

The authors emphasise that their results are relevant to a canonical circuit and describe the canonical circuit comprising three types of interneurons, however, their model only incorporates two types of interneurons. I understand this might be a work in progress. Nevertheless, I find it would be better to avoid mentioning the "canonical circuit" if the theory does not actually describe the canonical circuit.

Further, authors use the assumption that PV neurons are not tuned to the stimulus and justify this with findings of one empirical study. It seems rather that tuning of PV neurons is an open question, with some studies reporting relatively strong tuning of PV interneurons in the cortex (see e.g. Runyan et al., Neuron 2010,  Moore and Wehr, J. Neurosci. 2013). While it is justifiable to use the assumption of untuned PV interneurons, I find it better to present it as an assumption supported by some of the empirical literature.

**Questions:**

Unless I misunderstood something, the model is by design limited to encoding of a single stimulus feature. An increasing amount of literature in neuroscience shows that biological neural networks (in particular in the cortex), typically represent several stimulus- and behaviour-related variables (Or at least, such variables can be reliably decoded from neural activity in vivo; see for example Stringer et al. Nature 2019, Stronger et al. Science 2019, Mussall et al. Nature 2019). These observations include the same neuron being responsive to multiple features of the stimulus or of the behaviour. Can authors comment on that?

Does Bayesian sampling implemented by the model imply a static stimulus and a convergence of network activity to such stimulus? How does the network deal with a stimulus that has rapid shifts in dynamics or that changes the mean?

Can the model deal with non-Gaussian stimuli?

Tuning curves of neurons of the same cell type are homogeneous  across neurons (have the same shape). Is this assumption required for analytical computations? Could there be a computational reason that would instead allow heterogeneity of tuning across neurons of the same cell type?

**Limitations:**

Limitations are adequately addressed.

---

> ### Author Rebuttal · Authors · 2024-08-05
>
> We appreciate the reviewer's feedback and positive comments about math analysis, and the insight of SOM neuron's role in inference presented in our manuscript
>
> > The biggest weakness of the paper...
>
> Thanks for your suggestions; we will revise our manuscript accordingly by adding more content about intuitive descriptions. For example, we can add text related to Fig. 3C to explain why the structured inhibition from SOM neurons could speed up sampling by decreasing the temporal correlations of samples. Intuitively, Hamiltonian sampling comes with oscillation, and meanwhile, the structured inhibition from SOM will bring oscillations of stimulus samples computed by E neurons.
>
> > The authors emphasis that their results...
>
> Thank you for your suggestion. We will weaken our wording in the revision. We will just use the wording "canonical circuit" in describing the Fig. 1A and call our proposed model a circuit with diverse types of interneurons. In addition, we can also remove the "canonical" in the title.
>
> > Further, authors use the assumption that PV neurons...
>
> Thanks for the suggestion and we agree that the tuning strength of PV is under debate.  We will revise our text by emphasizing the untuned PV interneurons in the model is an __assumption__. This simplifies our theoretical analysis, while reserving a certain amount of biological plausibility, via multiplicative modulation on E tunings (Fig. 1G). Specifically, we will weaken our text in lines 104-105 into \
> "Some experiments found the stimulus orientation weakly modulates the PV neurons, hence, for simplicity, we assume PV neurons in the model are not tuned to stimulus features, providing only global unstructured inhibition to E neurons for stability (see Discussion for the complexity of PV neurons' tunings)"
> \
> In Discussion we will cite Runyan 2010 and Moore 2013 to discuss the experimental discovery of some PV neurons with sharp tuning, remind readers that the tuning of PV neurons is debatable, and clearly state that global inhibition from PV is an assumption considered in the study.
>
> > Unless I misunderstood something....
>
> Due to the page limit, our submitted manuscript only presents the 1D posterior sampling.  The proposed circuit with SOM neurons can sample multi-modal and high-dim posteriors. Please find the results in the rebuttal PDF and our response in Global Rebuttal.
> We are happy to include the results to strengthen our paper in the revised version with one more page.
>
> > Does Bayesian sampling...
>
> Yes, the current manuscript only considers a static stimulus and thus the stimulus samples embedded in the spatiotemporal neuronal activity will converge to the 1D stimulus posterior density, rather the just the point value of the stimulus.
>
> The proposed circuit model, with or without SOM neurons, can be used to infer the changing stimulus $z_t$, i.e., inferring the instantaneous posterior $p(z_t| r_t ... r_1)$  based on all history of neural inputs $(r_t, ..., r_1)$ in a hidden Markov model.
> Then the variance of the transition prior probability $p(z_t|z_{t-1})$ will be stored in the peak recurrent weight between E neurons, $w_{EE}$ (Eq. 2), decreasing the variance of $p(z_t|z_{t-1})$ (we have analytical results about this). This is because a larger $w_{EE}$ will incur a stronger bump response of E neurons (dark blue in Fig. 1E) which moves slower over the stimulus space, corresponding to a transition probability in MCMC with smaller variance.
>
> > Can the model deal with non-Gaussian stimuli?
>
> Yes, please see the attached Rebuttal PDF where the circuit can sample bimodal posteriors.
> In principle, the proposed framework can sample exponential family distributions (with a similar mechanism to Ma et al., Nat. Neurosci. 2006 and Zhang et al., Nat. Comms. 2023) or a mixture of exponential family distributions, where the __spatial profile__ of recurrent weights (Eq. 2) acts as the natural parameter of distributions and determines the type of sampling distribution in the circuit. The Gaussian distribution comes from the Gaussian profile of recurrent weights (Eq. 2).
> If we change the recurrent weight profile into other shapes, e.g., von Mises function, the circuit model can sample von Mises distributions.
>
> To sample distributions out of the exponential family, we may consider passing the responses of the current circuit model into a feedforward network which can rescale the Gaussian distribution into arbitrary distributions. This mechanism was used in a recent study, Chen et al., NeurIPS 2023 mentioned by Reviewer RifV.
>
> > Tuning curves of neurons of the same cell type are homogeneous...
>
> Yes, the homogeneity simplifies the math analysis significantly, without altering our conclusion.  Real cortical circuits have heterogeneity among neurons, and the homogeneity assumption has been widely used in many computational neuroscience models as a justifiable simplification, especially in the work of continuous attractor networks, e.g., Ben-Yi Shai 1995; Wu, Neural Computation 2008; Khona, Nat. Rev. Neurosci., 2022, etc.
>
> Heterogeneous neuronal tunings could be realized by introducing __randomness (zero mean with certain variance)__ into the recurrent connection matrix (Eq. 2), which has also been widely used in (chaotic) Excitation and Inhibition (E/I) balanced networks (Vreeswijk, Science 1996; Neural Computation 1998; Litwin-Kumar, Nat. Neurosci., 2012; Rosenbaum, Nat. Neurosci., 2017, etc).
>
> A potential __function of heterogeneity__ from random recurrent weights is that this puts the spiking networks into the chaotic regime where the network __internally generates Poisson variability__ (last term in Eq. 1). We think the injected multiplicative variability in our rate-based network (Eq. 1) captures the chaotic Poisson variability from spiking network dynamics. Then, stronger random recurrent weights will induce large heterogeneity, corresponding to a larger Fano factor $F_E$ of the injected variability in Eq. 1.

---

> > ### Comment · Reviewer_K6Di · 2024-08-10
> >
> > I thank the Authors for their reply.
> >
> > About the last point on heterogeneous tuning curves leading to chaotic dynamics, do I understand correctly that having heterogeneous tuning curves in the Eq.(1) instead of homogeneous ones, the stochastic term in the Eq.(1) could potentially be removed or replaced by a deterministic term?
> >
> > Unless I overlooked something, I find that the variable \xi (\theta,t) in the Eq.(1) is not clearly defined. Also, how does the type of stochastic process in the last term of Eq.(1) influence the type of sampling resulting from the network?

---

> > > ### Author Response · Authors · 2024-08-10
> > >
> > > Thanks very much for your reply!
> > >
> > > > About the last point on heterogeneous tuning curves leading to chaotic dynamics, do I understand correctly that having heterogeneous tuning curves in the Eq.(1) instead of homogeneous ones, the stochastic term in the Eq.(1) could potentially be removed or replaced by a deterministic term?
> > >
> > > Yes, this is exactly what we suggested.
> > >
> > > In computer simulations of sampling algorithms, we call a built-in __pseudo-random__ function whose underlying dynamics is also __chaotic__. Conceptually, this is similar to the chaotic E/I balanced network dynamics.
> > > In simulating a neural sampling network model, although we also call a random function and inject the variability into the model, we are concerned that real neural circuits probably do not have a statistically stable source of variability like the random function in programming. And it is less likely a stable variability can magically or trivially emerge in neural circuits.
> > > Therefore we think how the noise (or variability) used in sampling can be generated in neural circuits is an important issue at the neural implementation level, but which has not been paid sufficient attention in the field.
> > >
> > > Due to the page limit, we don't have sufficient space to emphasize the importance of the multiplicative variability (Eq. 1 last term) in our study. Please find our below reply.
> > >
> > > > Unless I overlooked something, I find that the variable \xi (\theta,t) in the Eq.(1) is not clearly defined. Also, how does the type of stochastic process in the last term of Eq.(1) influence the type of sampling resulting from the network?
> > >
> > > Sorry for the confusion. The $\xi(\theta, t)$ is a standard Gaussian white noise with zero mean and unit variance, and satisfying
> > > $\langle \xi(\theta, t)\xi(\theta', t') \rangle = \delta(\theta - \theta') \delta(t - t')$.
> > > In the rate-based network model, we use the (continuous) Gaussian white noise to approximate the multiplicative Poisson variability that results from stochastic spike generation (lines 83-87). This approximation is reflected by that the variance of $\xi(\theta,t)$ is proportional to instantaneous synaptic input $u_E(\theta,t)$ (negatively rectified with $[\cdot]_+$ in Eq. 1).
> > >
> > > This __multiplicative__ variability is essential for the network model with __fixed weights__ to flexibly sample posteriors with different uncertainties (Fig. 2G, 3F and Eq. 21; Please check the significance of this function in Global Rebuttal and our reply to Reviewer __RifV__). That is, if we replace the multiplicative noise with an additive noise, e.g., $\sigma \xi(\theta, t)$, our network model with fixed weights cannot sample posteriors with different uncertainties.
> > >
> > > To intuitively explain the underlying mechanism, let's review the Langevin sampling dynamics (a copy of Eq. S33)
> > > $$\frac{dz}{dt} = \tau_L^{-1} \nabla \log p(z) + (\tau_L/2)^{-1/2} $$
> > > which requires the __drift and diffusion terms both share the same factor $\tau_L$__. How this stringent requirement could be achieved in a recurrent neural circuit model has not been investigated before, to our best knowledge.
> > > With the multiplicative noise in Eq. (1), the samples generated by $E$ neurons are governed by Eq. (13) (copied below),
> > >
> > > $$ \frac{dz_E}{dt} = \tau_E^{-1} g_{EF}(\mu_z - z_E) + \sigma_E \tau_E ^{-1/2} \xi_t $$
> > >
> > > The common factor in drift and diffusion terms is $\tau_E = \tau U_E$ (defined in line 185), which is proportional to $U_E$, the magnitude of the synaptic inputs of E neurons (defined in Eq. 9).
> > > Intuitively, this is because (the full math derivations are in Eqs. S23-S30)
> > >
> > > - Higher neuronal response (larger $U_E$) will have larger inertia, analogous to a heavier object having larger inertia in kinematics, which is reflected by a large value of $\tau_E$ in the drift term.
> > > - Due to the multiplicative noise (Eq. 1, last ter,), larger $U_E$ will incur larger nosies in single neurons. Then the projected noise on the $z$ dynamics (the low-dim attractor manifold) will be higher.
> > >
> > > Combined, the multiplicative noise effectively links the drift and diffusion coefficients in the Langevin dynamics to share a common factor.

---

> > > > ### Comment · Reviewer_K6Di · 2024-08-12
> > > >
> > > > I thank the Authors for their thorough reply.

---

### Official Review · Reviewer_NhnW · 2024-07-12

**Soundness:** 3
**Presentation:** 2
**Contribution:** 2
**Rating:** 6
**Confidence:** 2

**Summary:**

The paper proposes a dynamical model for how neurophysiologically realistic circuits of pyramidal excitatory neurons, with two types of inhibitory interneurons, could perform Bayesian inference in a generative model of a uniform prior (over a bounded domain such as orientation angles) and a Gaussian likelihood. Analyzing the proposed neuronal circuit, the paper finds that its dynamics implement Hamiltonian Monte Carlo, and that removing one type of inhibitory interneuron results in Langevin Monte Carlo.  The paper then claims to replicate certain neurophysiological findings, specifically regarding L2/3 of the laminar circuit but also, according to some experiments, of L5.

**Strengths:**

The paper's in-depth use of neurophysiology alongside dynamical systems is its greatest strength.

**Weaknesses:**

The paper is quite dense, often unclear for a machine-learning audience, and ultimately only addresses a uniform-Gaussian joint density that cannot capture natural stimuli or behavior.  However, the authors have addressed this weakness in their responses and demonstrated that their model can sample from multimodal distributions.

**Questions:**

Where do the authors propose that their circuit ought to be found, neuroanatomically?  How does it fit into the laminar structure proposed by existing cortical microcircuit theories?

**Limitations:**

See weaknesses.  The paper's chief limitation is considering an overly simplified generative model based on previous work on probabilistic population codes, neglecting the observation that priors can be decoded (under supervised learning settings and at above-chance rates, etc.) almost ubiquitously from cortical recordings and are modulated by top-down feedback activity.

---

> ### Author Rebuttal · Authors · 2024-08-05
>
> Thanks for the reviewer's appreciation of our math analysis and the biological plausibility of the model.
>
> > The paper is quite dense, often unclear for a machine-learning audience, and ultimately only addresses a uniform-Gaussian joint density that cannot capture natural stimuli or behavior.
>
> We will improve our manuscript to make it more friendly to machine learning audiences.
> We are sorry that the detailed justification of the biological plausibility of the circuit model and the theoretical analysis might appear less interesting to machine learning audiences, whereas they are probably necessary for a complete theoretical neuroscience study.
>
> The proposed circuit with SOM neurons can sample multi-modal and high-dim posteriors. Please find the results in the rebuttal PDF and our response in Global Rebuttal.
> We are happy to include the results to strengthen our paper in the revised version with one more page, if you encourage us to do that.
> The reason for not including them is because of the page limit and the completeness of the manuscript, where we sacrificed the inference task complexity to present the reasoning of building the model and math analysis.
>
> > Where do the authors propose that their circuit ought to be found, neuroanatomically? How does it fit into the laminar structure proposed by existing cortical microcircuit theories?
>
> The micro-circuit structure (Fig. 2A) is canonical and ubiquitous in all cortical brain regions in mammals. In recent years, most experiments utilized mice's visual cortex to investigate this canonical micro-circuit, because we have high-throughput recording and perturbing tools on mice.
>
> In terms of laminar layers, the canonical micro-circuit (Fig. 2A) typically exists in both layers 2/3 (superficial layer) and layer 5 (deep layer), although the connection strength among neurons varies across laminar layers and across brain regions. The Fig. 7 in Ref. [24] in the submitted manuscript is a good summary of this micro-circuit structure in both superficial and deep layers.
>
> > See weaknesses. The paper's chief limitation is considering an overly simplified generative model based on previous work on probabilistic population codes, neglecting the observation that priors can be decoded (under supervised learning settings and at above-chance rates, etc.) almost ubiquitously from cortical recordings and are modulated by top-down feedback activity.
>
> We agree that the __derived__ generative model considered in our theoretical neuroscience study is simple compared with machine learning studies.
> It's worth noting the difference in research philosophy of the presented study is that we didn't use a __top-down approach__ where we "assign" a simple 1D generative model and "design" a circuit model to implement the inference.
> Instead, we ask the question the other way around from a __bottom-up view__. That is, given the recurrent circuit models with types of interneurons that have been extensively studied in neuroscience in recent years (acknowledged by the Reviewer RifV by saying this is "the norm of papers in neural sampling theory"), what kind of generative model and probabilistic inference can be implemented by the biological circuit. This philosophy was detailed in Lines 139-143 and Lines 152-156 of the submitted manuscript.
>
> In this way, even if the discovered 1D sampling in the proposed circuit model is simple, we feel the result still has its merit to be reported, because
> 1) no previous study has proposed what kind of inference can be achieved in the circuit with types of interneurons;
> 2) the simple 1D inference will encourage people to further investigate how the canonical circuit could do more complex inference.
>
> For the results uploaded in the Global Rebuttal PDF, we found the circuit with SOM neurons could sample multi-modal stimulus. However, due to the page limit, we didn't include them in the submitted manuscript. And we are happy to include them in the revised manuscript if you would prefer.

---

> > ### Comment · Reviewer_NhnW · 2024-08-11
> >
> > My thanks to the authors for their additional analyses and results.  I agree with them that the bottom-up, neuroscience-first approach is important to this paper, and so am satisfied to have the additional results included in supplementary material for the final manuscript.
> >
> > However, a last request: could the authors please clarify throughout the manuscript when they are referring to a canonical intralaminar microcircuit vs the more typical usage of a canonical laminar (eg: across the column) microcircuit?  This clarification was important for my understanding of the whole paper and needs to come across in the text.
> >
> > From there I can see through to raising my score.

---

> > > ### Author Response · Authors · 2024-08-11
> > > **Cortical regardance of our circuit model**
> > >
> > > Thanks for your reply. We will improve the manuscript further by adding more background about the canonical circuit, interneurons, and laminar layers, in order to make our manuscript more friendly to machine learning readers.
> > >
> > > To explain the cortical regardance of our circuit model, let's talk about some background of the cerebral cortex. If we flatten the cerebral cortex, it is a 3D neural sheet with $x-y$ axes denoting the positions on the cortical surface, and the $z$ axis denoting the depth perpendicular to the cortical surface. If we move along the $x-y$ directions, we have different cortical regions, whereas moving along the $z$ direction, we go through 6 laminar layers.
> > >
> > > For the __cortical column__, it is regarded as the cylinder structure perpendicular to the cortical surface, i.e., the set of neurons spanning the whole $z$ cortical axis but only a tiny part along the $x-y$ axes. Typically, the neurons within a cortical column encode a similar stimulus feature, e.g., the neuronal column preferring a $0$ degree of moving direction. Typically there are hundreds of excitatory neurons (E) within a column. \
> > > Furthermore, a __cortical hypercolumn__ is regarded as a set of cortical columns (spanning more on $x-y$ axes compared with one column) containing a full set of preferred stimulus feature values, e.g., all columns preferring all moving directions.
> > > The size of the hypercolumn in the primary visual cortex is about $1$ mm $\times$ $1$ mm along the $x-y$ axes. \
> > > At last, a __cortical region__ is composed of many cortical hypercolumns. For example, each hypercolumn processes the visual inputs at a particular visual location by having all neurons prefer different moving directions.
> > >
> > > Therefore, our model is regarded as the microcircuit within one laminar layer of a cortical hypercolumn, rather than a model including interactions across laminar layers.
> > > We mostly regard our model as the layer 2/3 within a hypercolumn in the primary visual cortex, since most of the experimental evidence comes from there.
> > > Specifically, $u_E(\theta)$ in Eq. (1) in the paper regards as the excitatory (E) neurons in layer 2/3 of a cortical column preferring direction $\theta$, which mathematically captures the mean response of all E neurons there. Then $\{ u_E(\theta)\}_\theta$ captures the responses of E neurons of the whole hypercolumn.

---

### Official Review · Reviewer_6oXx · 2024-07-13

**Soundness:** 3
**Presentation:** 3
**Contribution:** 2
**Rating:** 5
**Confidence:** 3

**Summary:**

This paper studied how the introduction of two inhibitory neuron population affects Bayesian inference in firing rate models with additive noise terms. Analytical derivation and simulations are done with circular 1D input variable (orientation). They found the inclusion of SOM leads to faster inference.

**Strengths:**

1) I would like to thank the authors for their clear writing and detailed technical reports;
2) there are many decision points in building up the circuit model, e.g. firing rate vs. spiking model, choice of weight structure etc., all decisions are carefully thought of and reasoned in the paper, leading to a very technically solid paper.

**Weaknesses:**

My main reservation with the paper is on its significance - I think the paper (at least in the way it is written) interests people within the specific subfield of Bayesian sampling in simplified circuit models for its technical merits; the result of SOM leads to faster inference for simple 1D input may interest SOM researchers in neuroscience. I'm just debating how much of these merit a NIPS publication.

**Questions:**

Have the authors considered having a mixture of two orientations as input and study how the introduction of I-diversity resolves the potential interference? (mentioning it not to ask for more experiments for this submission but just as a general future direction)

**Limitations:**

Limitations are adequately addressed.

---

> ### Author Rebuttal · Authors · 2024-08-05
>
> We appreciate the reviewer's positive feedback on our writing and math analysis!
>
> > My main reservation with the paper is on its significance - I think the paper (at least in the way it is written) interests people within the specific subfield of Bayesian sampling in simplified circuit models for its technical merits; the result of SOM leads to faster inference for simple 1D input may interest SOM researchers in neuroscience. I'm just debating how much of these merit a NIPS publication.
>
> Understanding how neural circuits in the brain do computation is one of NeurIPS's interests, which is also acknowledged by the Reviewer K6Di (strengths). We believe this adventure would bring mutual benefit to both neuroscience and machine learning (ML) communities, in that it not only helps us understand the brain, but also has the potential to be a new building block for ML tasks (Of course, our current model is still a "baby" and is not able to deal with real tasks).
> There are lots of famous examples of brain-inspired computations, e.g., Neocognition by Fukushima, Conv Net by Yan LeCun, etc.
>
> Even if the manuscript only presents a 1D posterior sampling, its analytical methodology and research philosophy (analytically identified circuit representation and algorithm) further our understanding of the __uncertainty representation__ in neural networks, and how different types of interneurons contribute to sampling, which, we believe, will have long-term impacts on ML. In contrast, conventional perceptron-based neural network models ignored neural fluctuations when they were developed decades ago, which causes __uncertainty representation__ in artificial neural networks, a fundamental question in ML.
> Moreover, the diversity of interneuron types may provide new possibilities to implement/link with the complicated components used in conventional RNNs used in ML such as LSTM.
> At last, the 1D sampling is the preface of our series of work, and we have also considered extending the circuit model to deal with more complicated posteriors (please see our Global Rebuttal and Rebuttal Figure).
>
> > Have the authors considered having a mixture of two orientations as input and study how the introduction of I-diversity resolves the potential interference? (mentioning it not to ask for more experiments for this submission but just as a general future direction)
>
> Yes, the proposed circuit with SOM neurons can sample multi-modal and high-dim posteriors. Please find the results in the rebuttal PDF and our response in Global Rebuttal.
> We are happy to include the results to strengthen our paper in the revised version with one more page, if you encourage us to do so.
> The reason for not including them is because of the page limit and the completeness of the manuscript, where we sacrificed the inference task complexity to present the reasoning of building the model and math analysis.

---

> > ### Comment · Reviewer_6oXx · 2024-08-12
> >
> > The added results are important to be included in the main text. Assuming they will be included, I increase my score.

---

> > > ### Author Response · Authors · 2024-08-12
> > >
> > > Thanks very much for your positive reply!
> > > We will include the new results in the main text of the revised manuscript.

---

### Author Rebuttal · Authors · 2024-08-05

## Global rebuttal
We thank all reviewers' efforts in reading our manuscript!

### Neural circuit sampling of complex posteriors
Three Reviewers (6xXx, K6Di, and RifV) wondered whether the circuit model can sample more complex posteriors, e.g., multi-modal and/or high-dim posteriors. Yes, the proposed circuit model framework can sample these complex posteriors (see the attached rebuttal PDF file), including
- Multi-modal posteriors (Fig. 5). When simultaneously presenting two stimuli to the proposed circuit model (Fig. 5B), the circuit with SOM neurons can sample bimodal posteriors. In contrast, without SOM, the circuit only samples an unimodal distribution corresponding to the Gaussian approximation of the bimodal posterior.
- High-dim posteriors (Fig. 6). To sample high-dim posteriors, we can extend the current model into many coupled networks with each the same as the current manuscript. Each network receives a neural input generated from a latent 1D stimulus, and samples the corresponding 1D latent stimulus. Combining the samples generated by all networks, the coupled circuit as a whole samples high-dim posteriors with the dimension determined by the number of networks. The coupling weights between networks store a high-dim correlation prior of stimuli (Fig. 6C). In this way, the coupled networks are similar to the network presented in Zhang et al., Nat. Comms. 2023 (see its Fig. 6), but each subnetwork in Zhang's study didn't consider SOM neurons.

__We are happy to include the new results in the final version as long as reviewers think they can strengthen our manuscript__.

Due to the page limit, we didn't include them in the submitted manuscript. Even for the 1D posterior sampling, the Reviewer Nhnw feels the manuscript is __dense__.  To fit the page limit, the submitted manuscript sacrificed the complexity of inference tasks for presenting our reasonings for crafting a circuit model with sufficient biological plausibility (lines 67-133, Fig. 1G-H and Fig 4C-D; acknowledged by the Reviewer 6xXx), and rigorously analytical results of the nonlinear neural dynamics (Eqs. 9-21; acknowledged by all Reviewers). We feel the __biological plausiblity__ and the __analytical results__ are just as important as the complexity of inference tasks, especially since no previous work investigated inference in circuit models with different types of interneurons.
Certainly, we are happy to revise it based on reviewers' suggestions.

### The simplicity of 1D posterior sampling in the circuit model
We agree with reviewers that 1D sampling is computationally simple, however, its implementation in __nonlinear__ neural circuits with types of __interneurons__ remains unclear.
In addition, it's worth noting that the difference in the research philosophy of the present study is that we didn't use a __top-down approach__ where we "assign" this simple 1D generative model and "design" a circuit model implementation.
Instead, we ask the question the other way around from a __bottom-up view__: given the circuit models with types of interneurons that have been extensively studied in neuroscience (acknowledged by the Reviewer RifV, "the norm of papers in neural sampling theory"), what kind of generative model and probabilistic inference can be implemented by the biological circuit (detailed in Lines 139-143 and Lines 152-156 in submitted manuscript).
In line with this philosophy, we feel the current result still has its merit to be reported, because
1) no previous study has hypothesized what kind of inference can be achieved in the circuit with types of interneurons;
2) the simple 1D inference will stimulate people to investigate more complex inferences in the canonical circuit (shown in the Rebuttal PDF).

### The significance of the present study, and comparison with earlier studies
- The present study is probably one of the first one linking 1) a canonical nonlinear circuit model with __interneurons__, 2) __analytical__ results of circuit dynamics (Eqs. 11-12), and 3) Bayesian __inference algorithms__ (Eqs. 13 and 17) together.
\
__Comparison__: Although some earlier work studied more complicated inferences than ours, they __sacrificed the complexity/biological plausibility/theoretical analyses__ of the circuit model. For example, they either considered a linear/simplified neural model sacrificing the biological plausibility (e.g., Hoyer, NIPS 2003; Savin, NIPS 2014; Aitchison, Plos Comp. Bio. 2016), or studied a nonlinear, biologically plausible dynamics but without analytical results of the nonlinear dynamics (e.g., Benjamin, ICLR 2024; Chen, NeurIPS 2023; Echeveste, Nat. Neurosci., 2020). In comparison, our work analyzes the nonlinear circuit dynamics and analytically identifies the circuit algorithm, which is also acknowledged by all Reviewers.

- The circuit model with __fixed__ weights can flexibly sample posteriors with __different uncertainties__ (Fig. 2G & 3F).
\
This is a very important requirement because real neural circuits sample different posteriors within the time scale of hundreds of milliseconds which are too short to change synaptic weights.
\
__Comparison with earlier work__:
This flexible computation requirement has not been paid sufficient attention in many earlier studies, and thus we are unsure whether they could achieve this or not, including, e.g., Chen NeurIPS 2023 and Benjamin, ICLR 2024 (mentioned by Reviewer RifV); Dong, NeurIPS 2022; Savin NIPS 2014; Aitchison, Plos Comp. Bio. 2016; Hoyer, NIPS 2003.

- The injected __multiplicative__ variability is biologically plausible in that the neuronal responses are (Poisson-like)__, i.e., the variance of neuron's responses is proportional to its mean firing rate (Fig. 1G-H, the shared region increases with mean firing rate). The multiplicative noise is necessary for the circuit model with fixed weights to sample posteriors with different uncertainties.

We hope our reply will address the reviewers' concerns and reassess our work.

---

### Decision · Program_Chairs · 2024-09-25

**Decision:**

Accept (poster)

**Comment:**

This paper considers how canonical cortical circuits with multiple interneuron types can implement sampling algorithms. The reviewers broadly agreed a strength of this paper is in its tight connection between the physiology and the sampling algorithm, which will be of interest to the Bayesian inference community. In light of this, I recommend acceptance.

There are some concerns that the paper only presents simple posteriors. I recommend the authors utilize the extra page allotted for accepted papers to include results with more complex priors.